# THE HUMAN BRAIN AS A DYNAMIC MIXTURE OF EXPERT MODELS IN VIDEO UNDERSTANDING

**Christina Sartzetaki**[*1], **Anne W. Zonneveld**[*1], **Pablo Oyarzo**[2], **Alessandro T. Gifford**[2], **Radoslaw M. Cichy**[2], **Pascal Mettes**[1], **Iris I.A. Groen**[1]

[1]Informatics Institute, University of Amsterdam, The Netherlands
[2]Department of Education and Psychology, Freie Universität Berlin, Germany
{c.sartzetaki, a.w.zonneveld, i.i.a.groen}@uva.nl

## ABSTRACT

The human brain is the most efficient and versatile system for processing dynamic visual input. By comparing representations from deep video models to brain activity, we can gain insights into mechanistic solutions for effective video processing, important to better understand the brain and to build better models. Current works in model-brain alignment primarily focus on fMRI measurements, leaving open questions about fine-grained dynamic processing. Here, we introduce the first large-scale model benchmarking on alignment to dynamic electroencephalography (EEG) recordings of short natural videos. We analyze 100+ models across the axes of temporal integration, classification task, architecture, and pretraining, using our proposed *Cross-Temporal Representational Similarity Analysis (CT-RSA)* which matches the best time-unfolded model features to dynamically evolving brain responses, distilling $10^7$ alignment scores. Our findings reveal novel insights on how continuous visual input is integrated in the brain, beyond the standard temporal processing hierarchy from low to high-level representations. After initial alignment to hierarchical static object processing, responses in posterior electrodes best align to mid-level temporally-integrative action features, showing high temporal correspondence to feature timings. In contrast, responses in frontal electrodes best align with high-level static action representations and show no temporal correspondence to the video. Additionally, temporally-integrating state-space models show superior alignment to intermediate posterior activity, in which self-supervised pretraining is also beneficial. We draw a metaphor to a dynamic mixture of expert models for the changing neural preference in tasks and temporal integration reflected in the alignment to different model types across time. We posit that a single best-aligned model would need such training and architecture as to allow combining and dynamically switching between these capacities.

## 1 INTRODUCTION

Humans are able to perceive and understand a highly dynamic world efficiently, with neural representations changing dynamically in response to continuous visual information. The framework of representational alignment (Sucholutsky et al., 2025) provides a rich resource to investigate how humans achieve this and for guiding model design. Within computational cognitive neuroscience, this framework is used to identify the mechanisms giving rise to cognition through hypothesis testing with task-performing computational models (Kriegeskorte & Douglas, 2018; Doerig et al., 2023), with deep neural network computer vision models being the best models capturing neural responses in visual cortex (Yamins et al., 2014; Güçlü & Van Gerven, 2015). Conversely, machine learning can draw from cognitive neuroscience to inform more efficient and robust human-like artificial intelligence, e.g. through implementations of brain-like energy constraints (Lu et al., 2025a), feedback pathways (Konkle & Alvarez, 2023), human-like development (Lu et al., 2025b), perceptual straightening (Hénaff et al., 2019; Bagad & Zisserman, 2025), or even explicit finetuning on neural or cognitive data (Muttenthaler et al., 2025; Safarani et al., 2021; Moussa et al., 2025). Computational cognitive neuroscience has employed large-scale benchmarking as a tool for systematic and reproducible comparisons of model-brain representational alignment (Conwell et al., 2024) under natural

---

*These authors contributed equally to this work.

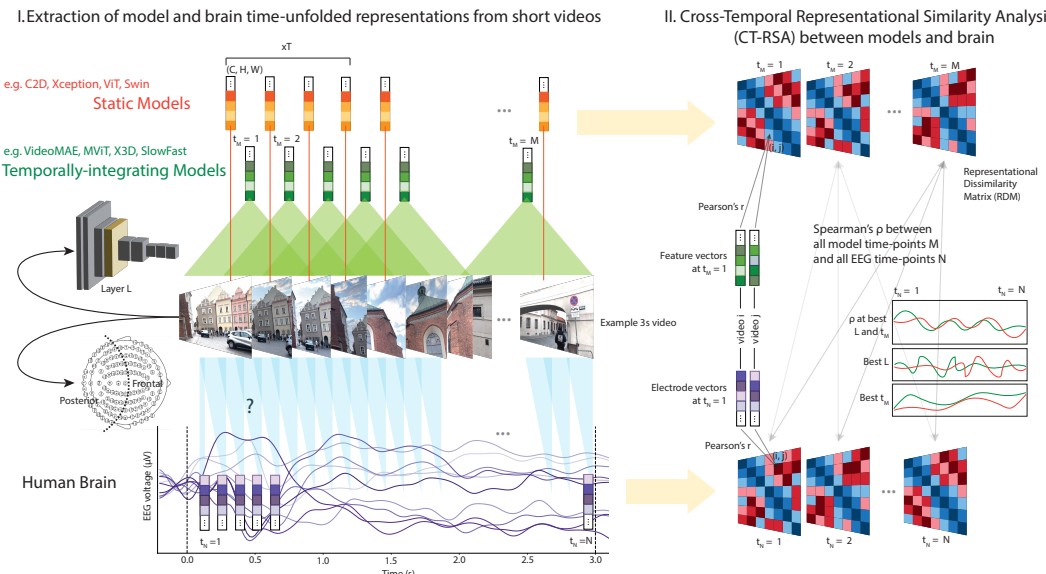

Figure 1: Our method for evaluating alignment of 100+ models with dynamic EEG responses to video. The extraction of time-unfolded representations from both systems (left) is followed by CT-RSA (right), which computes the maximum model-brain alignment across all timepoints and layers of a model. We systematically vary the axes of temporal integration, classification task, architecture, and pretraining - here we highlight differences in model temporal integration.

- mostly static image - stimuli, following the introduction of neural benchmark datasets (Schrimpf et al., 2018; Cichy et al., 2019; 2021; Gifford et al., 2023; 2024; Allen et al., 2022; Hebart et al., 2023; Lahner et al., 2024). However, static images lack temporal context, which strongly affects visual processing (Willems & Peelen, 2021), as for example illustrated by the temporal adaptation of neural responses (Benda, 2021; Brands et al., 2024). Critically, findings from static image perception do not automatically generalize to real-world conditions (Russ et al., 2023; David et al., 2004). This highlights a need for large-scale benchmark initiatives for dynamic video stimuli.

Recent work benchmarked model alignment to video fMRI (Sartzetaki et al., 2025; Garcia et al., 2025; Tang et al., 2025), showing how key axes of variation in those models (e.g. architecture, task training, temporal integration) affect brain alignment. However, these works did not yet consider two crucial factors: the rich dynamics that characterize neural responses, accessible non-invasively in humans via M/EEG, as well as the dynamics of temporally unfolded model features. *In this study, we employ large-scale alignment benchmarking of deep neural network models on dynamic human brain responses to natural videos using EEG, and leverage unfolded model features through our proposed Cross-Temporal Representational Similarity Analysis* (CT-RSA). **Our contributions:**

- We present the first large-scale representational alignment benchmarking on dynamic brain responses (EEG) to natural videos, evaluating 100+ static and temporally-integrating deep neural network models using Cross-Temporal Representational Similarity Analysis (CT-RSA), which matches the best time-unfolded model features across the EEG time-course.
- Our results reveal a changing neural preference for semantic tasks and temporal integration over time, and preferential specialization across the brain hierarchy (posterior-frontal). Posterior activity evolves from alignment with static, hierarchically increasing object processing to mid-level action representations, continuously tracking the video through temporal integration - whereas frontal activity is best captured by early, static semantic action representations.
- Leveraging the time-unfolded model features, we show a strong temporal correspondence between model time and EEG time in posterior activity, but not in frontal activity.
- Comparing the additional axes of variation of model architecture and pretraining within models of the same task and temporal integration, we show that state-space models better align to intermediate posterior processing, in which self-supervised pretraining is also beneficial.

Overall, our results suggest that neural representations reflected in fine-grained dynamic brain measurements are not best captured by any single DNN model type but rather resemble a dynamic mixture of expert models that allows switching between semantic tasks and temporal integration, revealing opportunities for brain-inspired representation learning on videos.

## 2 RELATED WORK

**Dynamic brain measurements under static and dynamic natural stimuli.** Human brain dynamics are studied at different timescales with distinct methods. For short timescales, usage of M/EEG is essential to make millisecond-level inferences, whereas the sluggish hemodynamic response underlying fMRI prevents such precision. For static stimuli (i.e., images), findings from M/EEG studies of brain dynamics show that discriminable representations emerge as quick as within 50 ms after stimulus onset (Cichy et al., 2014), due to a fast feedforward sweep of visual information processing from early to later stage brain areas, i.e. the cortical hierarchy (Serre et al., 2007). Early neural processing of static images has been found to reflect low-level visual features (Groen et al., 2013), while later neural processing to reflect high-level representations, e.g. categorical processing (Cichy et al., 2014) and task performance (VanRullen & Thorpe, 2001), illustrating a gradual emergence of high-level representations over time: a "temporal hierarchy"(Bankson et al., 2018; Cichy et al., 2016). This has not only been observed for static objects, but also for static scenes (Greene & Hansen, 2020) and static actions (Zimmermann et al., 2025). From these studies, high-level action information appears to emerge later (250 ms) than high-level object and scene information (150 ms). This may reflect a need for integration of contextual information for action recognition (El-Sourani et al., 2018), with mid-level features like motion patterns, objects, and scene information being processed in parallel before being integrated at the level of action semantics (Zimmermann et al., 2025).

Human brain dynamics in response to short dynamic stimuli (i.e., videos) are comparatively much less explored. Using fMRI Lahner et al. (2024) found a hierarchical correspondence between action recognition model layers and brain regions as in static images, also demonstrating different levels of sensitivity to frame-shuffling and neural temporal correspondence to the video especially in early visual cortex (EVC) - consistent with prior findings on the hierarchy of temporal receptive windows (Hasson et al., 2008; Lerner et al., 2011). Additionally, work by Jung et al. (2025) showed that high-level semantic action features uniquely explain activity over a widespread range of cortex during dynamic natural vision, more so than semantic agent or scene features. A few works have used EEG to investigate brain dynamics to natural videos using domain-specific social interaction videos; Dima et al. (2022) found a temporal hierarchy of visual, action-related and social-affective features, while McMahon et al. (2025) showed that mid and high-level social features are decodable at similar timings, using EEG-fMRI fusion. Using engine-rendered (naturalistic) short videos, Karapetian et al. (2025) also found a temporal processing hierarchy and revealed that motion and action-related features are processed faster during video than static frame presentation. However, large-scale natural video EEG datasets have been missing so far, limiting research on the fine-grained temporal dynamics of real-world visual processing. *In this work,* we are diving in underexplored territory by leveraging a newly collected, large-scale high temporal resolution EEG dataset for domain-general (i.e. not social-specific) and natural (i.e. not rendered) dynamic stimuli.

**Benchmarking model representational alignment on short-video fMRI.** Very recent works have focused on large-scale benchmarking of model-brain alignment for videos. Garcia et al. (2025) used fMRI of short social interaction videos, comparing 8 video action recognition models with 348 image object recognition models. They found that image models captured representations in EVC and the ventral stream, whereas video models outperformed in the lateral stream, linked to social cognition. Sartzetaki et al. (2025), used fMRI of short domain-general videos, comparing 47 video models, 41 image object recognition models and 11 image action recognition models to disentangle temporal integration from classification task effects. Temporally-integrating models surpassed static models in EVC, through better alignment of middle model layers, while action recognition models surpassed object ones in later stage brain regions via classification layers. Tang et al. (2025), used 5 different video fMRI datasets and compared 92 models, including image object recognition models, video action recognition models, multimodal models, and non-NN baselines. Through correlating brain alignment with model zero-shot performance in different tasks, they found that appearance-free (motion-only) action recognition and object recognition are the two most relevant tasks for brain alignment, with task-agnostic self-supervised models performing best at both tasks and alignment. Current studies are however all based on fMRI, precluding conclusions about fine-grained temporal dynamics. Similarly on the model side, the temporally-evolving internal feature dimension remains unexploited. *In this work,* we present the first large-scale investigation of model alignment with highly dynamic brain representations recorded via EEG during short video viewing. Building on the model set and axes of variation from Sartzetaki et al. (2025), we compare how different model groups and their internal temporal representations align with the brain across processing time.

## 3 METHODOLOGY

Figure 1 shows an overview of our methodology for measuring alignment of models to dynamic brain responses[1]. We base our methodology on Sartzetaki et al. (2025) but extend it in: (1) applying it to an EEG instead of fMRI dataset, (2) expanding the model architectures sampled, (3) unfolding the temporal dimension of the extracted model features and (4) proposing a temporal extension to the representational alignment metric. We describe these aspects in the next three sections.

### 3.1 CROSS-TEMPORAL REPRESENTATIONAL SIMILARITY ANALYSIS (CT-RSA)

Neural network models compute representations at subsampled frame intervals, while the mapping of specific frames onto brain responses is uncertain, raising the question of how to align model and brain representations over time. We extend Representational Similarity Analysis (RSA) (Kriegeskorte et al., 2008) to compare all temporally unfolded model features with all EEG timepoints without imposing assumptions about the relationship between the two, performing a cross-temporal matching that identifies the maximally aligning model timepoint at each EEG timepoint. Such cross-temporal matching also allows studying dynamic prediction (de Vries & Wurm, 2023) and temporal generalization (King & Dehaene, 2014). We choose to build on RSA to perform a multivariate analysis that benchmarks emergent brain alignment of model representations. [2]

**Temporal Representational Dissimilarity Matrices. (A) EEG.** At each EEG timepoint $t_N$ we create a super-subject brain Representational Dissimilarity Matrix (RDM) ($B_{t_N}$), using a particular electrode subset with repetition-average channel vectors $v_{s-t_N}$ for each participant (subject) $s$. We compute the super-subject RDM by averaging the RDMs of all subjects at that timepoint, as $B_{t_N} = avg_s(B_{s-t_N})$, $B_{s-t_N}^{ij} = 1 - r(v_{s-t_N}^i, v_{s-t_N}^j)$, $\forall i, j(i < j)$, $0 \leq j < K, 0 \leq t_N < N$ where $K$ is the total number of videos and $r$ is Pearson correlation. **(B) Models.** We flatten the features of each model layer $l$ and timepoint $t_M$ into a one-dimensional feature vector of length CxHxW, and reduce its dimensionality using Sparse Random Projection followed by Principal Component Analysis to 100 Principal Components to obtain $f_{l-t_M}$. We compute one RDM per model layer and model time-point as $M_{l-t_M}^{ij} = 1 - r(f_{l-t_M}^i, f_{l-t_M}^j)$, $\forall i, j(i < j)$, $0 \leq j < K$, $0 \leq t_M < M$.

**Cross-Temporal correlation of RDMs.** To calculate the cross-temporal representational alignment score between the model and EEG timecourses of RDMs, we first compute Spearman's $\rho$ for all combinations of model and EEG timepoints for each layer $R_{l-t_M t_N} = \rho(B_{t_N}, M_{l-t_M})$. We then choose the highest-correlating model layer and model timepoint per EEG timepoint $R_{t_N} = max_{l-t_M}(R_{l-t_M t_N})$. This maximization is essential to retrieve insights from approximately 1k RSA scores that result from all combinations of model layers and timepoints. Over all EEG timepoints, electrode subsets, and models, this allows us to handle over $10^7$ RSA scores.

**Noise ceiling computation.** Because of individual subject variability in brain data, noise ceilings for each electrode subset are computed to compare model RSA scores against the maximum obtainable score given the inter-subject variability, following standard approaches in the field (Nili et al., 2014). For the lower noise ceiling (LNC) we compute a mean RDM across all subjects except one. Then we take the Spearman correlation of the left-out subject RDM and mean RDM, repeat for all the subjects and calculate the average. For the upper noise ceiling (UNC) we take the mean of all RDMs without removing subjects, compute the Spearman correlation of each subject RDM with the mean RDM, and average. The UNC signifies perfect correlation for the amount of noise in the data, often referred to as the maximum amount of variance that can be explained. All further reported alignment scores are scaled by the UNC, and so is the LNC.

**Statistical significance.** To test if model RDMs correlated significantly with the brain RDMs, we performed permutation tests (Nili et al., 2014), in similar fashion as done in Sartzetaki et al. (2025). For each model RDM at every EEG timepoint we select the model RDM for highest-correlating layer and model timepoint pair and permute rows and columns 10000 times using the same 10000 random permutations. We then calculate a null distribution per EEG timepoint by computing the Spearman correlations of all permuted RDMs with the brain RDM. For significance of a group of

---

[1]Due to licensing restrictions of the stimuli used in the brain dataset, the video frames shown in the figure are sourced from representative videos captured by the authors themselves and are not subject to copyright.

[2]We incorporate CT-RSA in the Net2Brain python library (Bersch et al., 2025), see recent release at `https://github.com/cvai-roig-lab/Net2Brain`.

models against zero, we perform a two-tailed sign test between the average null distribution of all models in the group and the average observed Spearman correlation, corrected by subtracting the average observed Spearman correlation before stimulus onset, to account for inflated pre-stimulus correlation scores arising from the maximization inherent to our CT-RSA method. To test for significant differences between two groups of models, we perform a two-tailed sign test between the null distribution created from the across-group differences in the within-group average distributions, and the observed difference in the average of the two model groups' Spearman correlations. When showing group medians instead of means, the above statistical inferences are performed on the medians. All statistical inferences are corrected for multiple comparisons across time points using FDR correction (Benjamini & Hochberg, 1995) with a cluster threshold of two consecutive time points.

## 3.2 VIDEO MODELS

We expand upon the original set of 99 video models as used in Sartzetaki et al. (2025) to additionally reflect changes in the current state-of-the-art, such as the introduction of state-space models (SSMs) as a scalable linear-complexity alternative to Transformers (Gu & Dao, 2024), for both static vision (Zhu et al., 2024) and video (Li et al., 2024). With the addition of 3 VisionMamba object recognition and 8 VideoMamba action recognition models, our final set of models includes 44 image models trained for object recognition on ImageNet, 10 image models trained for action recognition on Kinetics 400, and 49 video models trained for action recognition on Kinetics 400. Image action models are trained on videos but treat time in a trivial way (separate computations per frame). An additional set of 7 models trained for action recognition on other datasets (Kinetics 710 and Something-Something-v2) was also evaluated, bringing the total to 110 computer vision models tested overall. Comprehensive and extended lists of models can be found in Appendix B.

**Temporally unfolded feature extraction.** Since the models process a fixed number of frames at a given sampling rate, each video is divided into S sub-clips of length T, differing per specific model. For each layer, we extract features of shape (T, C, H, W) from each sub-clip, and unfold the temporal dimension to obtain TxS=M model timepoint features of shape (C, H, W). We extract features from all higher-level blocks in the models and include the final classification layer.

## 3.3 EEG DATASET

We utilize the newly collected EEG Moments Dataset (EEGMD), an EEG-based extension of the large scale video fMRI BOLD Moments Dataset (BMD) (Lahner et al., 2024). EEGMD covers the exact same set of 1102 short (3s) natural videos from Moments in Time (Monfort et al., 2019; 2021). See Appendix A for more information on EEG acquisition. The dataset consists of EEG recordings of 6 participants for a "train set" of 1000 videos with 6 repeats per video and a "test set" of 102 videos with 24 repeats, using a set-up of 128 electrodes. Data was recorded at a sampling rate of 1000 Hz with online filtering (between 0.1 Hz and 100 Hz) and rereferenced (to Fz). In the current analysis we use the test set for the application of RSA, as signal to noise ratio is higher due the greater number of stimulus repetitions to average across (see decoding results in Appendix C Fig. 6), similar to Sartzetaki et al. (2025). The test set videos are representative of the whole dataset, covering a wide range of objects, actions and scenes (for more elaborate description of the video contents see Lahner et al. (2024)). All main analyses are based on the group-average level; we provide additional supplementary results for subject level analyses in Appendix C Fig. 9.

**Preprocessing and electrode selections** We performed offline preprocessing using Python and MNE (Gramfort et al., 2013). The continuous EEG data was epoched into trials from -0.2s to 3.5s with respect to stimulus onset, baseline-corrected by subtracting the mean of the pre-stimulus period separately for each trial and channel, and temporally downsampled to 50 Hz. Next, we performed multivariate noise normalisation (MVNN) based on the covariance matrices of each timepoint to reweigh (un)reliable sensors and to (de)emphasise specific spatial frequencies, as recommended for multivariate analyses such as RSA (Guggenmos et al., 2018). Our analyses are based on two electrode partitions. First, we select posterior electrodes (35), which overlay visual cortex and are commonly used in vision studies (Xie et al., 2020; Seijdel et al., 2021; Loke et al., 2024). To make a coarse distinction between the opposite ends of the brain's spatial organization, we select a second partition of frontal electrodes (54), covering (pre)frontal cortex associated with executive functions. The inclusion of frontal electrodes is an exploratory choice, further motivated by the lack of prior knowledge on the localization of visual processing during longer timescales (video). See Appendix A for the names of all included electrodes.

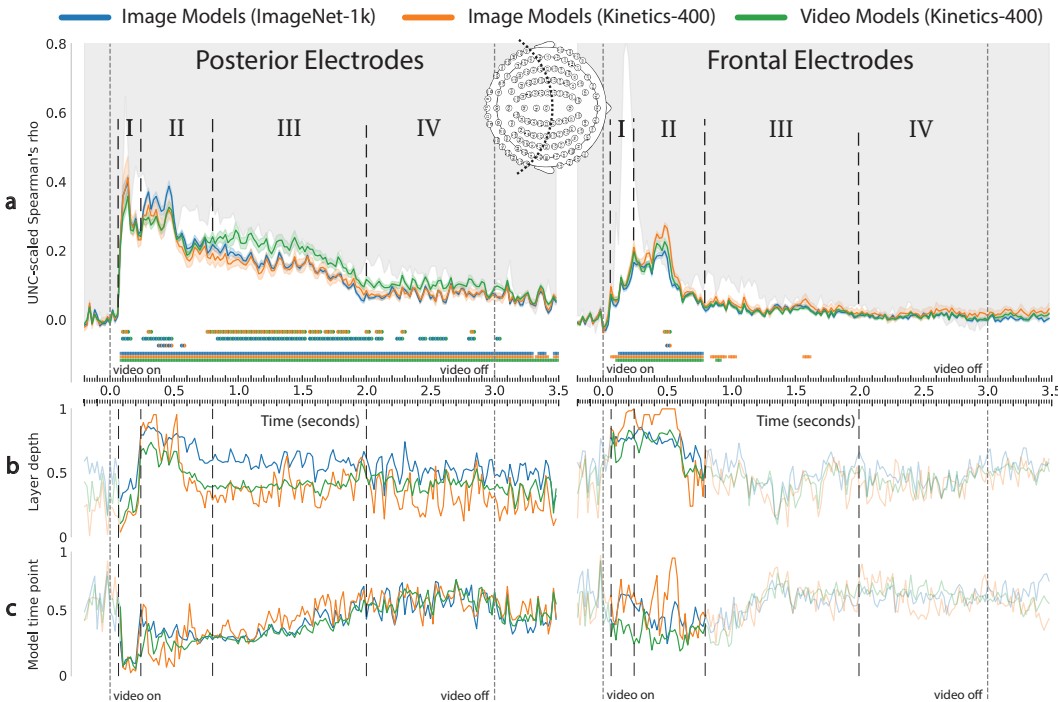

Figure 2: (Left) Task and temporal integration types dynamically interchange in posterior electrodes. (Right) Action task specificity is early and stable in frontal electrodes. Row (a) shows the maximum score over all model timepoints and layers (group average), with significance marked by squares (against zero) and two-colored circles (pairwise), while (b) and (c) show the layer and timepoint that yield the scores. Processing stages I-IV are identified and LNCs are outlined in gray background.

## 4 RESULTS

We assess brain alignment over time for 110 deep neural networks using CT-RSA, considering four main axes of variation: temporal integration, classification task, architecture, and pretraining.

**Task and temporal integration types dynamically interchange in posterior electrodes.** In Fig. 2 (Left: Posterior) we compare models across the whole EEG time-course by varying (1) the temporal integration, i.e. image (static) v.s. video (temporally-integrating) models, and (2) the classification task, i.e. object (ImageNet-1k) v.s. action recognition (Kinetics-400) models. RSA scores (Fig. 2a) in all model groups are significant against zero from 0.08s, extending to the duration of the whole video and offset. Based on visual inspection of the development of these scores, we distinguish four temporal stages of processing: (I) 0.06s - 0.24s, (II) 0.24s - 0.8s, (III) 0.8s - 2s and (IV) 2s - 3s. To summarize the results of these different stages and explore model variation, we additionally evaluate bin-averaged scores in Fig. 3. We next discuss these stages in turn, simultaneously interpreting the score (Fig. 2a/3a) and the best model layer (Fig. 2b/3b) that gives rise to it. First in **stage I,** scores peak for all model groups at 0.14s and image models significantly outperform video models, with the best model being AlexNet. For all model groups this high correspondence can be accounted for by relatively early model layers (below 0.5), and as the peak in scores decays, by progressively later layers. This suggests that during this stage, posterior processing reflects static low-level information, independent of task. In **stage II,** object recognition models show a clear peak, outperforming the other model groups; the best model here is a DenseNet. In contrast with stage I, the significant scores for all model groups are due to late layers, primarily relating posterior processing at this stage to static high-level object information. During **stage III,** we observe a steady decrease in image model scores, while the score for video models increases and then remains relatively stable, before dropping towards the end of the stage. This leads to video models outranking the other groups (best model being MViT-v2), with scores being driven by mid-level layers. Thus, overall stage III can be summarized as a mid-level temporally-integrative action processing stage. During **stage IV**, scores and layer depth across all model groups remain stable after reaching their lowest point, with video models less significantly outperforming image models than in stage III. Looking across stages in Fig. 2b/3b, action recognition model scores correspond to an earlier layer than object recognition ones in stages I and III, with less clear patterns in the other stages.

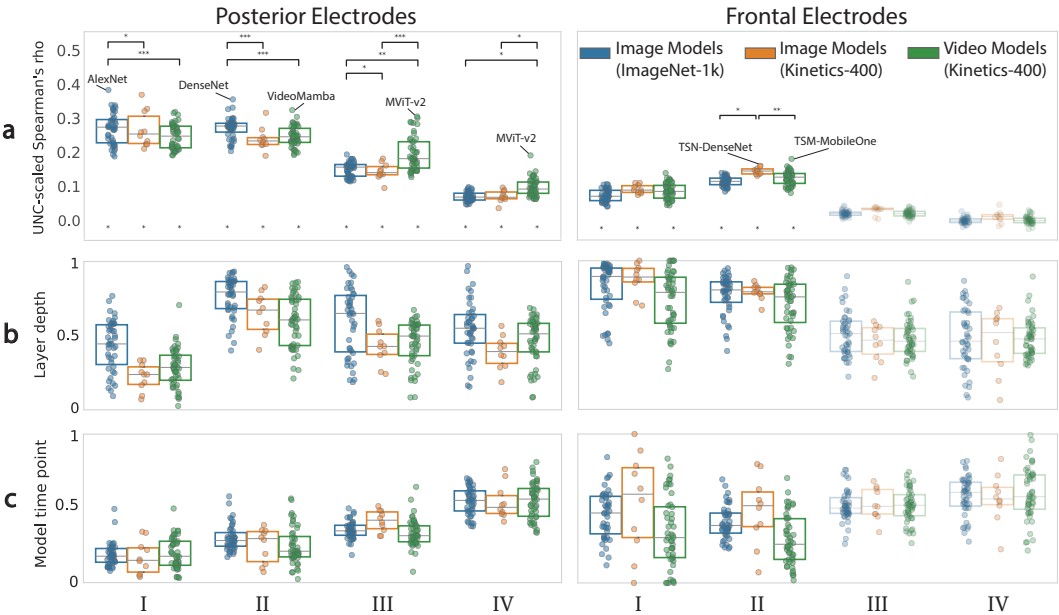

Figure 3: Bin-averaged results showing model variability within each of the temporal processing stages in Fig. 2, outlining best-aligned models in row (a). Significance is marked by a star at $y = 0$ (against zero) and by the respective number of stars for $p < 0.05, 0.01,$ and $0.001$ on top (pairwise).

The best model timepoint (Fig. 2c/3c) reveals a strong correspondence between model and EEG timepoints in posterior electrodes: early EEG timepoints best correspond to early model timepoints (close to 0), and later EEG timepoints to gradually later model timepoints (yet only up to ∼0.6), highlighting the temporality of posterior neural responses. Note also that where layer depth is high (stage II), the match in model timepoint is less strict.

The analysis demonstrates that posterior processing during video perception is highly temporal, progressing through distinct stages that emphasize different representational demands. An early peak around 0.14s reflects low to mid-level static processing, followed by a rapid shift toward higher-level static object representations. Beyond this point, posterior parts of the brain gradually transition into mid-level temporally-integrative action processing, marking the late stages of video perception.

**Action task specificity is early and stable in frontal electrodes.** In Fig. 2/3 (Right: Frontal) we show results for the frontal partition. We observe that most neural processing of video information occurs in stages I and II, as model groups show no significant alignment scores afterwards. Scores in this period show two peaks for all model groups, at 0.24s and 0.5s, with static action recognition models significantly outperforming the other groups at the last peak. This results from correspondence with late model layers. Regarding the best model timepoint (Fig. 2c/3c) and in contrast to posterior electrodes, we see that in frontal electrodes, during the period of significant scores, there is no clear temporal correspondence between model and EEG timepoints - with large spread across models (Fig. 3c). These results suggest that during video perception, neural processing reflected in frontal electrodes shows limited temporality and is primarily engaged until 0.8s with high-level, action-related information that is largely independent of within-video dynamics.

**Context window and sampling rate influence alignment in late posterior processing.** In Fig. 8 we show the effects of model context window length (num. of frames) and sampling rate (fps) on brain alignment. We observe a significant positive correlation between these factors and posterior brain alignment for temporally integrating models during stages III and IV, during which temporally integrating models also show the highest alignment (Fig. 2a/3a). These findings reinforce the notion that late-stage posterior processing of videos relies on temporal integration, with larger temporal context windows and higher temporal resolution enhancing alignment.

**State-space video models best capture mid-level intermediate posterior processing.** In Fig. 4 we compare brain-alignment of different architectures, i.e. CNNs, Transformers, or SSMs, focusing on the stages I-IV identified in the previous sections. To avoid confounds from task or temporal integration, we focus this analysis on video action recognition models. SSMs are the most brain-aligned to posterior channels in stages I-II, especially in stage II, and through earlier layers (mid-depth).

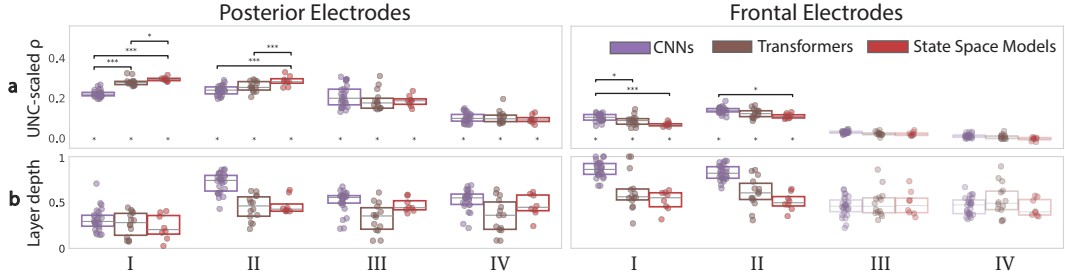

Figure 4: Architecture variation in bin-averaged scores for each stage. (Left) Video SSMs best capture mid-level intermediate posterior processing. (Right) CNNs give a slight edge in alignment to high-level frontal processing.

Additionally in stage I, Transformers are significantly better aligned than CNNs. In frontal channels CNNs are most brain-aligned in stages I-II via late layers, however, only with marginal difference to Transformers. In stages III and IV of both channel sets differences are mostly not significant. In App. C we show additional comparisons that a) control for model pretraining (Fig. 10), and b) compare between object recognition instead of video action recognition models (Fig. 12). The effects of architecture are stable across pretraining types, but differences between object recognition models are more muted. Specifically, static object SSMs do not show the observed advantage for video SSMs in posterior stage II. This suggests that SSMs capture a distinct component of neural processing in posterior channels during phase II, previously linked to static high-level object representation, through temporally-integrative mid-level action features. Overall, these results show a disadvantage of CNNs in early processing and an advantage of SSMs in intermediate posterior processing. Considering their architectural differences to Transformers, the CNN disadvantage suggests global attention is useful for early alignment, while the SSM advantage suggests recurrent processing benefits alignment to subsequent intermediate responses. This latter advantage appears more related to recurrency across time than across patches, as for static SSMs it does not persist.

**Self-supervised to no pretraining switch in posterior electrodes.** In Fig. 5 we compare the brain alignment of video action recognition models having different types of pretraining, either no pretraining, supervised pretraining on images (object recognition), supervised pretraining on videos (action recognition), or self-supervised pretraining on videos. We observe that in stages I-II of the posterior electrodes self-supervised pretraining achieves superior brain alignment. In stage I it is on par with pretraining on image object recognition, while in stage II it is superior to all other types. In stage III, no pretraining is significantly better than all other types. In stage IV in posterior and in all stages in frontal electrodes, significant differences are less consistent, with only a slight advantage of supervised video pretraining in frontal. We again further control for the model architecture in App. C Fig. 13, and find that the advantages of self-supervised pretraining in stage II and no pretraining in stage III are robust across architectures. A conclusion we could draw from this analysis, is that the advantage of self-supervised pretraining in the primarily "object processing" stage relates to self-supervision enabling generalization to other tasks (through task-general pretraining objectives like spatiotemporal patch reconstruction), while the benefit of no pretraining in the temporally-integrative stage reflects avoiding the shortcut learning of unrelated patterns that is common during pretraining (Byvshev et al., 2022). We additionally compare action recognition fine-tuning datasets (Sth-Sth-v2, K710), showing no effects on alignment at any stage or partition (App. C Figs. 14-15)).

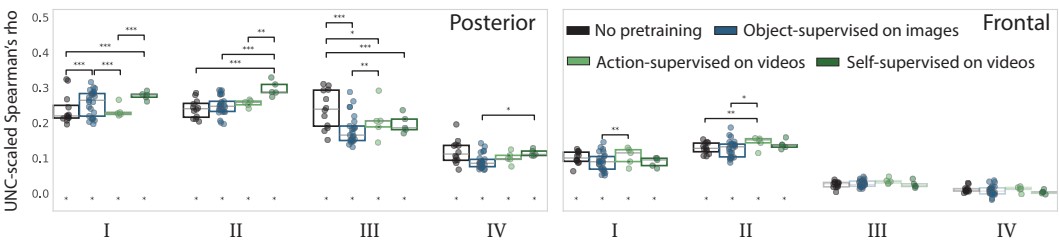

Figure 5: Pretraining variation in bin-averaged scores for each stage. (Left) Switch of benefit from self-supervised (stage II) to no pretraining (stage III) in posterior electrodes. (Right) Slight advantage of supervised video pretraining in frontal electrodes.

## 5    DISCUSSION AND CONCLUSIONS

In this work we performed a large-scale model comparison of both static and temporally-integrating deep neural networks to dynamic EEG recordings. Using CT-RSA we make the following main observations: (1) Neural processing during video perception in posterior parts of the brain is highly temporal and unfolds in distinct stages, best captured by different type of representations: from an initial alignment with static low-level features, to mid- and high-level object features, and finally to mid-level temporally integrative action features. (2) In contrast, neural processing during video perception in frontal parts of the brain shows restricted temporality and is best aligned with static high-level action features early in the video. (3) Architecture-wise, temporally-integrating action SSMs best capture mid-level representations in posterior electrodes, reflecting a different component of the response in the primarily object-related stage, while indicating potential benefits of global attention and recurrent temporal processing for alignment to early-intermediate responses. (4) Self-supervised pretraining helps capture brain representations in the early, primarily object-related stage, while performing no pretraining captures brain representations of later more temporally-integrative stages, indicating independent benefits of general pretext tasks and unbiased target domain features.

**What we learn by moving to dynamic natural stimuli.** Moving from static images to more real world dynamic videos, we find similarities and differences in neural processing between these two formats. Congruent with the works of Bankson et al. (2018), Greene & Hansen (2020) and Zimmermann et al. (2025) in the domain of image perception, we find a temporal hierarchy of object and action features in posterior processing, but only within the duration of stages I and II, i.e. the first 500 ms of processing, consistent with the stimulus durations in those studies. Our findings are also largely consistent with prior work on the neural dynamics of short videos showing a temporal hierarchy of (social) action-related features (Dima et al., 2022; McMahon et al., 2025; Karapetian et al., 2025), where low and high level object features emerge earlier than action features. However, as we extend beyond the timescales used in these studies (beyond 1s) we discover that mid-level temporally-integrative action features are most brain-aligned until the end of the video, challenging the notion of a strict temporal hierarchy. The critical role of temporal integration during this late-stage processing is further highlighted by the observed increase in alignment with larger context windows and higher temporal resolution. Notably, this sustained alignment occurs consequently after the peak in alignment with regards to static high-level action features in frontal areas, raising the possibility that feedback from frontal to posterior regions may contribute to shaping later stages of visual processing of dynamic action information. Multiple studies have posed that during object recognition in static context feedback information from prefrontal cortex to posterior regions is critical in shaping behaviorally sufficient object representations, especially under challenging conditions (Goddard et al., 2016; Kar & DiCarlo, 2021; Oyarzo et al., 2025). Specifically, Oyarzo et al. (2025) pose frontal cortex plays a corrective role under these conditions, reshaping high-level visual representations to resolve ambiguity. If we extrapolate, video processing can be viewed as a more demanding extension of static image processing. This increased complexity may explain the recruitment of frontal activity, which appears to contribute early on, followed by a reconfiguration of representations in posterior activity for updated mid-level feature processing.

**What we learn from using dynamic brain measurements for benchmarking.** Compared to prior work on video fMRI alignment benchmarking (Sartzetaki et al., 2025), which found high alignment of temporally-integrating video models through mid-layers in EVC, we find an advantage for these model features now temporally localized in stages III and IV, i.e. from 0.8s and onwards, in the posterior part of the brain, following initial alignment with high-level object model layers. Combining the insights from both studies, we hypothesize that mid-level dynamic processing could be performed by the EVC after the initial encoding of the video appearance. The high alignment of action recognition models reported in high-level visual cortex by Sartzetaki et al. (2025) was also observed in the current study, but in the frontal electrode partition, likely reflecting an even higher-level cortical stage of processing, during stage II (0.24s–0.8 s). Additionally, leveraging the temporal resolution of EEG, we find that object representations also contribute to video processing in the posterior part of the brain, being the most predictive during early processing but rather briefly, before the processing of dynamic features. A hypothesis for the absence of object contribution in fMRI alignment for video is that, relative to the whole video time-course, object features are processed only briefly, and this transient signal is lost in the aggregation of the fMRI response. Similarly, the enhanced alignment with static processing during stage I (0.06s-0.024s) in posterior electrodes may be too brief to appear in the fMRI signal. We can also compare our results for the other factors

of variation that study explored, namely architecture (without SSMs) and training dataset. For architecture they showed CNNs and Transformers being overall equivalent in fMRI, with differences only in which layer was best aligned; in CNNs the best aligning layer was a later layer ($\approx$0.6) than in Transformers ($\approx$0.2), primarily in EVC. Here we mostly find the same; CNNs and Transformers align overall similarly apart from posterior stage I which shows a strong overall advantage for Transfomers, and in posterior stages II, III, IV (especially II), the best aligning layer is a later layer in CNNs than in Transformers. In our results, this trend also continues in frontal electrodes. For training dataset, the fMRI study showed a disadvantage of ego-centric datasets in alignment to face-processing brain regions, whereas we find no differences in alignment to EEG (Appendix C Fig. 14), most probably due to the lack of fine-grained spatial resolution.

**Prospects for model building.** Our findings suggest that during video perception the brain performs computations that are not best represented by any single model during the entire EEG time-course, but rather by alternating semantic tasks and temporal integration strategies, resembling a dynamic mixture of expert models. This type of neural processing is also corroborated by studies of non-visual, higher cognitive systems like reasoning and the multiple demand system, with Mixture of Experts (MoE) models matching established brain pathways (Cook et al., 2025; AlKhamissi et al., 2025). We propose (1) that a single model could be best aligned to the whole time-course if it was trained on a sufficiently general objective (e.g. self-supervised masked modeling) so that it can develop experts for object and action recognition, and for temporally-integrating vs. static processing (2) that dynamically switching between those experts is a design choice with potential for human-like capabilities (e.g. efficiency). Regarding (1), preliminary results on a single model (VideoMamba, App. C Fig. 16) show that pure self-supervision does yield the highest alignment in stages I and II (potentially due to the general objective, as in Tang et al. (2025)), but is surpassed by the fine-tuned version at later processing stages, suggesting a further need for supervised training. Concerning (2), this dynamic switching might prove limited in block-based (i.e. CNN/Transformer) video models, as they process videos in fixed temporal blocks; any dynamic switching could then only occur within these time spans. Architectures better suited to support this property are recurrent neural networks (e.g. SSMs) as they process input sequentially, continuously updating their hidden states. Dynamic switching could then emerge implicitly or be enforced, potentially reducing computational costs via keeping overall network activation low, or using feedback connections to make use of prior experts. This exemplifies the potential of brain alignment for model building, inviting the machine learning and cognitive neuroscience communities to think again in a more unified way through the search of common core principles in both engineering and biological solutions.

**Application prospects of CT-RSA.** The usage of CT-RSA is not limited to the current setting. In principle, the method generalizes to any setting that involves comparing two forms of temporal, or rather sequential, representations. On the model side, this could be multimodal audiovisual and language representations across time (e.g. Gifford et al. (2024)), either from separate unimodal DNNs or multimodal DNNs, supporting analyses on when exactly in time multimodally integrative neural processing occurs rather than unimodal processing. On the neural side, it could likewise extend to long-format fMRI time series or other sequential neural measurements. For example in the set up of d'Ascoli et al. (2025), encoding could be replaced with CT-RSA as an option for exploring the inherent temporal model-brain alignment instead of maximizing neural predictivity. Related forms of cross-temporal matching have been used to study dynamic prediction (de Vries & Wurm, 2023) and temporal generalization (King & Dehaene, 2014) of representations.

**Limitations and future work.** While our setup involved 100+ models, extending Sartzetaki et al. (2025), specific model types should be investigated in more depth, such as purely self-supervised models and models with different types of recurrent mechanisms (SSMs, LRUs, RNNs). Additionally, fine-grained differences between video models can be further explored by using, instead of a coarse top-down grouping, a bottom-up grouping that focuses on variations in temporal feature generalization. Another exciting direction is collecting neural data with intermediate duration videos (10-20s) that could include elements of visual surprise, camera changes, or scene cuts. Lastly, given the limited spatial resolution of EEG, our ability to draw conclusions about the precise spatiotemporal dynamics is constrained. To more directly assess whether, when and how the prefrontal cortex could be involved in the neural processing of videos, a promising direction for future work would be to employ model based EEG–fMRI fusion (Hebart et al., 2018; Cichy & Oliva, 2020; Cichy et al., 2014), combining the current EEGMD with BMD (Lahner et al., 2024) and leveraging the the complementary spatial and temporal resolution of these different brain measurements.

ACKNOWLEDGMENTS

CS is supported by an ELLIS Amsterdam Unit grant to IIAG. AWZ acknowledges support from the UvA Data Science Centre, as part of the Human Aligned Video AI Lab.

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

APPENDIX

## A EEG DATASET DETAILS

The EEG Moments Dataset was collected by the Neural Dynamics of Visual Cognition group, at the Freie Universität Berlin. The dataset will be made public in the near future.[3] EEG data during natural video viewing was collected for 6 participants, all with normal or corrected-to-normal vision. Procedures were approved by the ethical committee of the Department of Education and Psychology at Freie Universität Berlin and were in accordance with the Declaration of Helsinki. Stimuli were the exact same as in Lahner et al. (2024): 1102 3-second second videos, sampled from Monfort et al. (2019; 2021). Stimuli were square-cropped and resized to 268x268 pixels (5°×5° visual angle).

Stimuli were divided in non-overlapping sets, a "train set" of 1000 videos with 6 repeats and a "test set" of 102 videos with 24 repeats. The dataset is accompanied with crowd-sourced metadata, to annotate each clip with five word-level scene, object, and action labels, sampled from Places365 (Zhou et al., 2017), THINGS (Hebart et al., 2023), and Multi-Moments in Time (Monfort et al., 2021) respectively. Additionally, provided were five sentence-level text descriptions, a spoken transcript, a memorability score and a memorability decay rate.

The experiment consisted of a passive viewing paradigm with an orthogonal detection task to have participants stay attentive, where participants had to report after X nr of random trials whether a specific object/scene/action was present. The total experiment consisted of 8 sessions, that each contained 3 repeats of the 250 train videos and 3 repeats of the 102 test videos. Each session consisted of 16 runs of 66 trials, in between which participants could take breaks.

Every trial sequence consisted of 1s of a blank baseline screen, followed by 3s showing a video, another 0.25s baseline and then 2s of blink time. In the case of prompted trials the second baseline was followed by a 3s prompt screen, which was then again followed by the 2s blink time. Stimuli were presented on a grey background with a red fixation cross present during the whole experiment.

EEG data was recorded using a 128-channel actiCAP set up with electrodees arranged according to the standard 10-10 system (Nuwer et al., 1998) and a Brainvision actiCHamp amplifier. Data was recorded at a sampling rate of 1000 Hz with online filtering (between 0.1 Hz and 100 Hz) and rereferenced (to Fz). We performed offline preprocessing using Python and MNE (Gramfort et al., 2013). The continuous EEG data was epoched into trials from -0.2s to 3.5s with respect to stimulus onset and subsequently baseline corrected by subtracting the mean of the pre-stimulus interval for each trial and channel separately. The data was then temporally down- sampled to 50 Hz. Next, we performed multivariate noise normalisation (MVNN) based on the covariance matrices of each time-point to reweigh (un)reliable sensors and to (de)emphasise specific spatial frequencies, as recommended for multivariate pattern analysis methods like RSA (Guggenmos et al., 2018). We performed analyses on two separate sets of electrodes: a posterior partition consisting of 35 electrodes and a frontal partition consisting of 54 electrodes. The posterior partition contained the following electrodes: 'Pz', 'P3', 'P7', 'O1', 'Oz', 'O2', 'P4', 'P8', 'P1', 'P5', 'PO7', 'PO3', 'POz', 'PO4', 'PO8', 'P6', 'P2', 'P9', 'PPO9h', 'PO9', 'O9', 'OI1h', 'PPO1h'. The frontal partition contained the following electrodes: 'Fp1', 'F3', 'F7', 'FT9', 'FC5', 'FC1', 'FT10', 'FC6', 'FC2', 'F4', 'F8', 'Fp2', 'AF7', 'AF3', 'AFz', 'F1', 'F5', 'FT7', 'FC3', 'FCz', 'FC4', 'FT8', 'F6', 'F2', 'AF4', 'AF8', 'F9', 'AFF1h', 'FFC1h', 'FFC5h', 'FTT7h', 'FCC3h', 'FCC4h', 'FTT8h', 'FFC6h', 'FFC2h', 'AFF2h', 'F10', 'AFp1', 'AFF5h', 'FFT9h', 'FFT7h', 'FFC3h', 'FCC1h', 'FCC5h', 'FTT9h', 'FTT10h', 'FCC6h', 'FCC2h', 'FFC4h', 'FFT8h', 'FFT10h', 'AFF6h'.

---

[3]For further inquiries or requests, contact Alessandro Gifford (alessandro.gifford@gmail.com)

# B  MODEL DETAILS

Table 1: Model families. Action recognition models are trained on Kinetics 400 (Kay et al., 2017); those also available on other datasets are marked by $a$,$b$.

| Image Object Recognition | | | | | | Action Recognition | | | | | |
|---|---|---|---|---|---|---|---|---|---|---|---|
| | CNNs | | Transformers | | SSMs | | CNNs | | Transformers | | SSMs |
| 1 | AlexNet | 2 | CAiT | 3 | VideoMamba | 6 | CSN | 2 | MViTv2$^b$ | 8 | VideoMamba |
| 2 | DenseNet | 2 | ConViT | | | 5 | I3D | 2 | TimeSformer | | |
| 2 | EfficientNet | 2 | DEiT | | | 1 | R2P1D | 2 | Uniformer | | |
| 2 | RegNet | 2 | MViTv2 | | | 2 | SlowFast | 2 | Uniformerv2$^a$ | | |
| 4 | ResNet | 3 | Swin | | | 4 | Slow$^a$ | 1 | VideoMAE | | |
| 2 | ResNeXt | 1 | Twins | | | 1 | TPN | 2 | VideoMAEv2 | | |
| 4 | VGG | 2 | ViT | | | 5 | TSM$^b$ | 3 | VideoSwin$^a$ | | |
| 2 | WideResNet | | | | | 2 | X3D | | | | |
| 1 | Inception | | | | | | | | | | |
| 2 | RepVGG | | | | | | | | | | |
| 2 | SeResNe(X)t | | | | | 4 | C2D | 1 | TimeSformer | | |
| 2 | Xception | | | | | 4 | TSN$^b$ | 1 | TSN | | |
| 26 | | 14 | | 3 | | 26+8 | | 14+2 | | 8 | |

$^a$Availability also on Kinetics 710 (Carreira et al., 2019)
$^b$Availability also on Something-Something-v2 (Goyal et al., 2017)

Table 2: Exhaustive account of all models.

| Image Object Recognition | | | Action Recognition | | |
|---|---|---|---|---|---|
| CNNs | Transformers | SSMs | CNNs | Transformers | SSMs |
| AlexNet | CAiT_S | VideoMamba_T | IR_CSN_R152 | MViTv2_S$^b$ | VideoMamba_T_IN1k_f16 |
| DenseNet161 | CAiT_XXS | VideoMamba_S | IR_CSN_R152_BNfrozen_IG65M | MViTv2_B$^b$ | VideoMamba_T_IN1k_f8 |
| DenseNet201 | ConViT_S | VideoMamba_M | IR_CSN_R50_BNfrozen_IG65M | TimeSformer_DivST | VideoMamba_S_IN1k_f16 |
| EfficientNetB3 | ConViT_B | | IR_CSN_R152_IG65M | TimeSformer_JointST | VideoMamba_S_IN1k_f8 |
| EfficientNetB6 | DEiT_S | | IP_CSN_R152_IG65M | Uniformer_S | VideoMamba_M_IN1k_f16 |
| RegNetX16gf | DEiT_B | | IP_CSN_R152 | Uniformer_B | VideoMamba_M_IN1k_f8 |
| RegNetY8gf | MViTv2_S | | I3D_R50 | Uniformerv2_B$^a$ | VideoMamba_M_mask_f16 |
| ResNet34 | MViTv2_B | | I3D_R50_dotprod | Uniformerv2_B_k710pre | VideoMamba_M_mask_f8 |
| ResNet50 | Swin_T | | I3D_R50_embgauss | VideoMAE_B | |
| ResNet101 | Swin_S | | I3D_R50_gauss | VideoMAEv2_S | |
| ResNet152 | Swin_B | | I3D_R50_heavy | VideoMAEv2_S | |
| ResNeXt50 | Twins_pcpvt_B | | R2P1D_R50 | VideoSwin_T | |
| ResNeXt101 | ViT_S | | SlowFast_R50 | VideoSwin_S$^a$ | |
| VGG11 | ViT_B | | SlowFast_R101 | VideoSwin_B | |
| VGG11BN | | | Slow_R50 | | |
| VGG19 | | | Slow_R101 | | |
| VGG19BN | | | Slow_R50_IN1k$^a$ | | |
| WideResNet50 | | | Slow_R50_IN1k_embgauss | | |
| WideResNet101 | | | TPN_R50 | | |
| InceptionV4 | | | TSM_R50$^b$ | | |
| RepVGGa2 | | | TSM_R50_dotprod | | |
| RepVGGb2 | | | TSM_R50_embgauss | | |
| SeResNet50 | | | TSM_R50_gauss | | |
| SeResNeXt50 | | | TSM_MobOne_s4 | | |
| Xception41 | | | X3D_S | | |
| Xception71 | | | X3D_M | | |
| | | | C2D_R50_nopool | TimeSformer_SpaceOnly | |
| | | | C2D_R101_nopool | TSN_Swin | |
| | | | C2D_R50_pool8 | | |
| | | | C2D_R50_pool16 | | |
| | | | TSN_R50$^b$ | | |
| | | | TSN_R101 | | |
| | | | TSN_D161 | | |
| | | | TSN_MobOne_s4 | | |

$^a$Availability also on Kinetics 710 (Carreira et al., 2019)
$^b$Availability also on Something-Something-v2 (Goyal et al., 2017)

## C  SUPPLEMENTARY ANALYSES

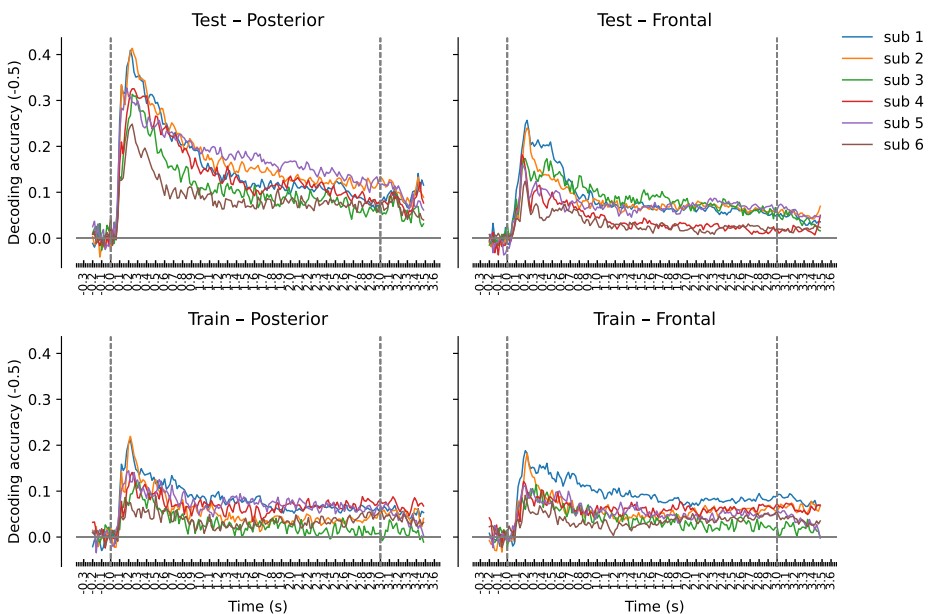

Figure 6: One versus one stimulus identity decoding per subject, corrected by chance level (-0.5). Note: the decoding accuracy on the train set was computed using a randomly selected subset of videos, matching the number in the test set (n=102).

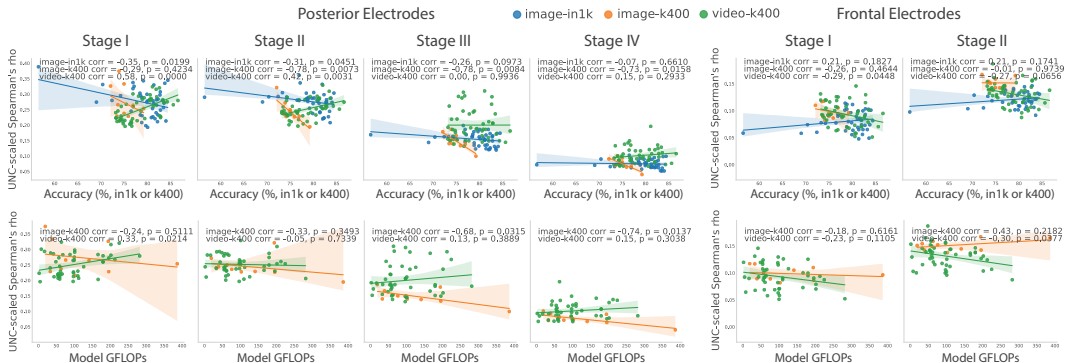

Figure 7: (Top) Relationship of brain alignment with model accuracy (in ImageNet-1k for object recognition models, and in Kinetics-400 for action recognition models). (Bottom) Relationship with model computational complexity (in GFLOPs). Only stages during which models showed significant brain alignment in Fig. 2 are shown.

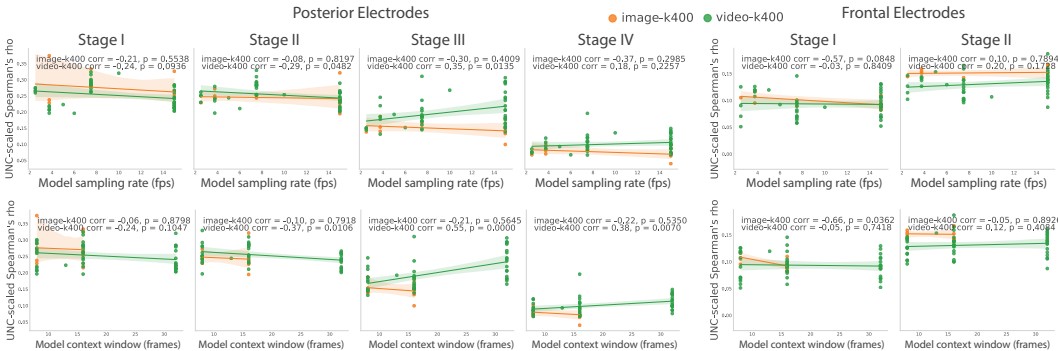

Figure 8: (Top) Relationship of brain alignment with model sampling rate (fps). (Bottom) Relationship with model context window (frames). Only stages during which models showed significant brain alignment in Fig. 2 are shown. Here, model sampling rate and context window positively correlate with the brain alignment score of temporally integrating models (video-k400) significantly (p<0.05) in posterior electrodes, stages III and IV.

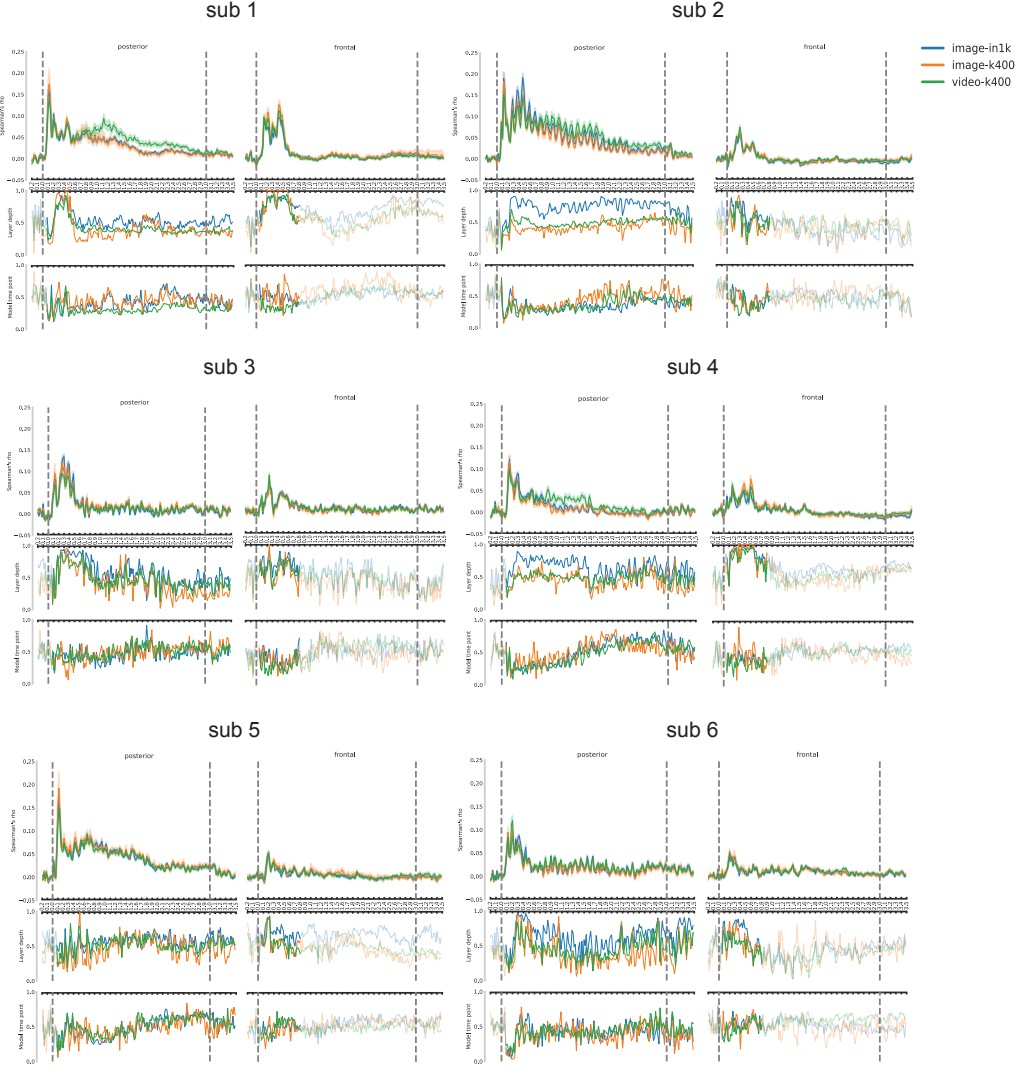

Figure 9: Subject level brain alignment results for the test set for posterior and frontal electrode partitions.

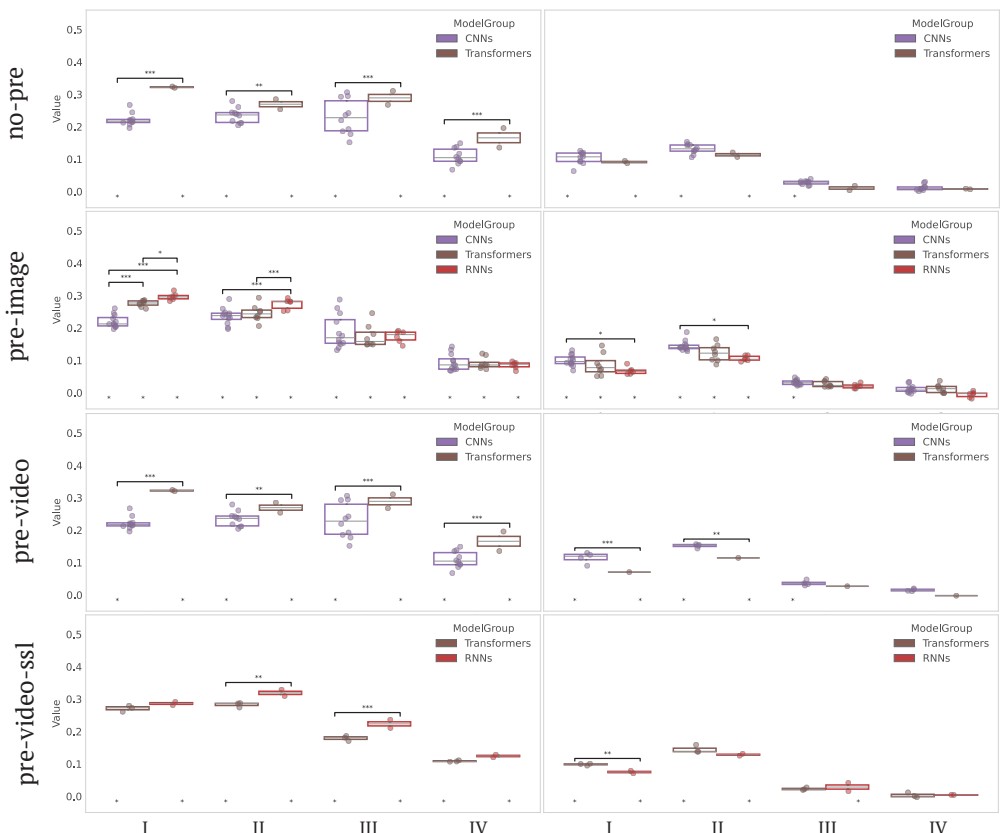

Figure 10: Additional control for pretraining in the architecture comparison of Fig. 4.

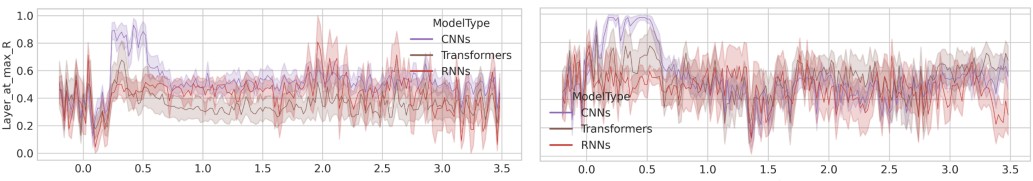

Figure 11: Showing the layer plot of Fig. 4b for the full timecourse.

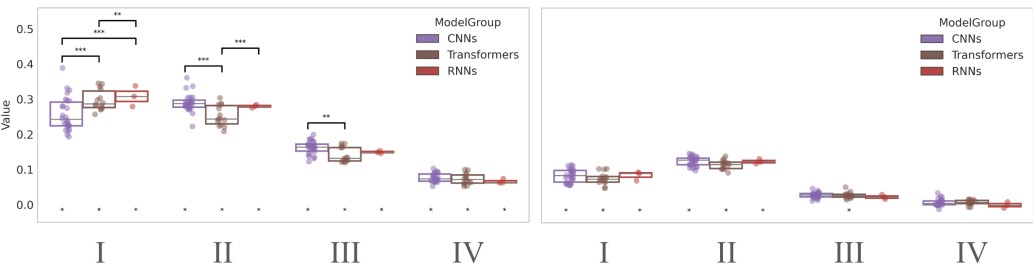

Figure 12: The comparison of Fig. 4 only now within object recognition models.

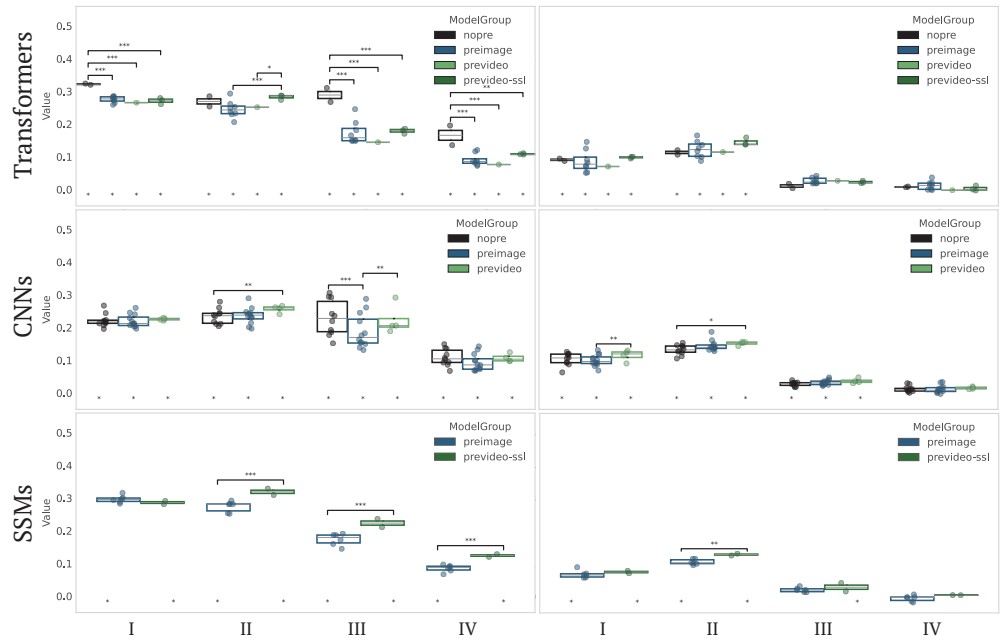

Figure 13: Additional control for architecture in the pretraining comparison of Fig. 5.

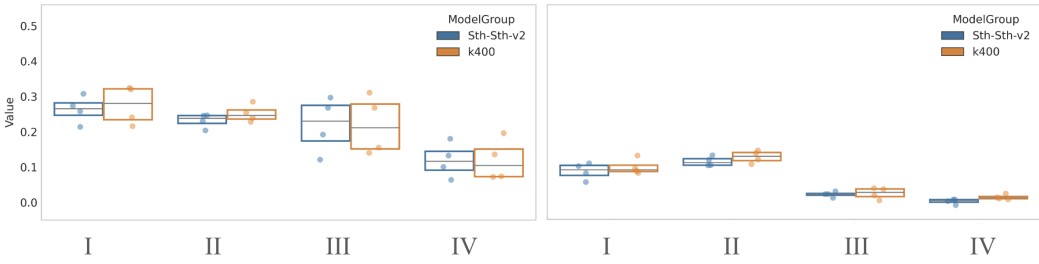

Figure 14: No significant differences between fine-tuning datasets (Sth-Sth-v2 vs. K400).

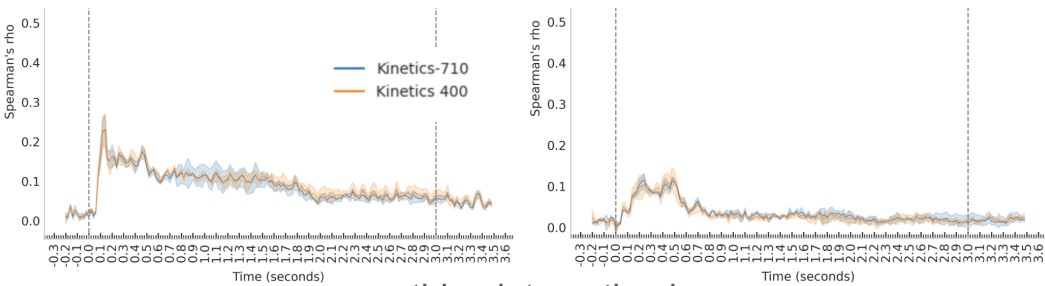

Figure 15: No significant differences between fine-tuning datasets (K710 vs. K400).

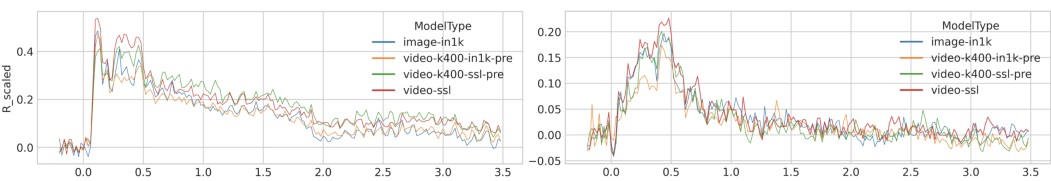

Figure 16: Comparing VideoMamba versions; object (blue), object-pretrained action (orange), self-supervised-pretrained action recognition (green), and pure self-supervised (red).

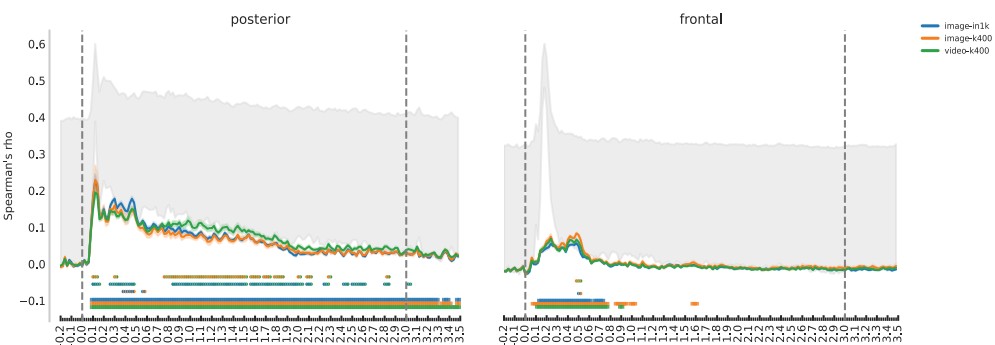

Figure 17: Same as Fig. 2 but without scaling to the upper noise ceiling. Gray shading shows both lower and upper noise ceilings.

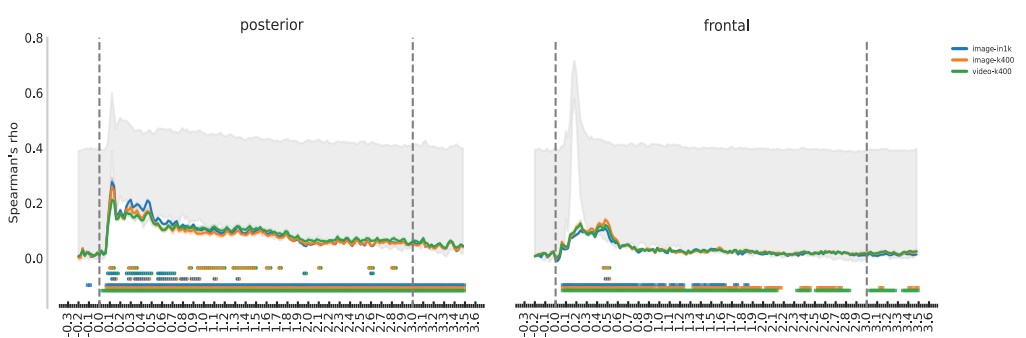

Figure 18: Non noise-scaled brain alignment of time-aggregated model features (RSA instead of CT-RSA). Gray shading portrays the noise ceilings.

## D    USE OF LARGE LANGUAGE MODELS (LLMS)

In the writing of this paper, LLMs were only minimally used to aid or polish writing, such as asking for synonyms or rephrasing at the sentence or sub-sentence level.

