# OpenReview forum: "The Human Brain as a Dynamic Mixture of Expert Models in Video Understanding"
_ICLR.cc/2026/Conference — ICLR 2026 Poster_

### Official Review · Reviewer_fzAq · 2025-10-31

**Soundness:** 3
**Presentation:** 3
**Contribution:** 3
**Rating:** 4
**Confidence:** 1

**Summary:**

This paper presents the first benchmark for the alignment between artificial image / video representations from neural networks and dynamic brain responses to natural videos, recorded using EEG.  This benchmark is carried out on a new dataset EEGMD (not public yet) that extends the Bold Moments Dataset (BMD) to EEG responses (to natural videos). To compare between models, the authors introduce a time-aware extension of the RSA method: Cross-Temporal Representational Similarity Analysis (CT-RSA), that allows computing dissimilarity between time-courses of EEG and model representational dissimilarity matrices. This allows them to make a number of significant findings about how the representations in artificial models and brains correlate across time and brain regions.

**Strengths:**

- The paper is well written, easy to follow, the methodology and metrics are clearly defined
- The assumption resulting from the analysis of the paper, that is the brain as a dynamic mixture of experts, is indeed supported by the results of the benchmark across the different architectures, time-aware / static models, model types, ...
- There are a number of conclusions that are new to the best of my knowledge (e.g. 'frontal activity being best capture by early static semantic action representations')

**Weaknesses:**

- The measurement of alignment is rather complex (UNC-scaled Spearman's rho) and the paper misses a discussion of why using this metric is superior to much simpler methods (e.g. just using Pearson's r across time). Since cluster analysis is absent, the advantage of using RDMs is not obvious.
- The paper makes a number of interesting findings but does not seem to delve into their interpretations (e.g. SSMs better aligning to posterior processing)

**Questions:**

- Apart from the already presented axis of analysis, could the authors present results correlating alignment with task-related performance of video / image embedding models on a given task ?

---

> ### Author Response · Authors · 2025-11-24
>
> Thank you for your review acknowledging our contributions, the novelty of our insights and support of our claims in evidence, as well as offering suggestions for the improvement of the manuscript. We have revised the manuscript and uploaded the new version with all changes relative to the original shown in blue. Our responses to your points one by one are as follows.
>
> ### W1: The measurement of alignment is rather complex (UNC-scaled Spearman's rho) and the paper misses a discussion of why using this metric is superior to much simpler methods (e.g. just using Pearson's r across time). Since cluster analysis is absent, the advantage of using RDMs is not obvious.
>
>
> The use of Spearman rather than Pearson correlation to compare model and brain RDMs is the most common and recommended practice in representational similarity analysis (Schütt et al. (2023)). Rank correlation is more general, as it does not assume a linear relationship. Crucially, it emphasizes capturing the relative structure of distances between RDMs rather than their absolute values. Furthermore, we scale each model’s Spearman correlation score by the upper noise ceiling (UNC), i.e. the average Spearman correlation of each subject RDM compared to the subject-mean RDM, to account for inter-subject variability, allowing us to interpret alignment relative to the maximum achievable agreement given variability across participants, as recommended by Nili et al. (2014). We additionally report unscaled results in Appendix C, Fig. 17 of the revised paper.
> We are unsure what ‘cluster analysis' in this context refers to. Could you please clarify?
>
> Schütt, H. H., Kipnis, A. D., Diedrichsen, J., & Kriegeskorte, N. (2023). Statistical inference on representational geometries. Elife, 12, e82566.
>
> ### W2: The paper makes a number of interesting findings but does not seem to delve into their interpretations (e.g. SSMs better aligning to posterior processing)
>
> We agree that our conclusions for the factors of architecture and pretraining were less clearly outlined in the paper than for the factors of classification task and temporal integration. We have now remedied that in the paper, and also reiterate here.
>
> For **architecture**, the most clear (based on significance) conclusion would be a disadvantage of CNNs and an advantage of SSMs in posterior phases I and II. Considering the differences of the two architectures, and at the same time their differences to Transformers, this might indicate that global attention as well as recurrent processing are useful for aligning to the initial cascade of information in the brain. This advantage appears to be more related to the recurrent processing across time rather than across space (since for static models it does not persist, see Appendix C, Fig. 12).
>
> For **pretraining type**, we hypothesize that the advantage of self-supervised pretraining in the primarily “object processing” stage could relate to self-supervision enabling generalization to other tasks, while the benefit of no pre-training in the temporally-integrative stage may reflect avoiding shortcut learning of unrelated patterns (Byvshev et al., 2022).

---

> > ### Author Response · Authors · 2025-11-24
> >
> > ### Q1: Apart from the already presented axis of analysis, could the authors present results correlating alignment with task-related performance of video / image embedding models on a given task ?
> >
> > Thank you for the question. We provide additional analyses in Appendix C, Fig. 7 (top) that show the **relationship of the model-brain alignment metric to model task performance**, in Kinetics-400 for the action recognition models and in Imagenet-1k for the object recognition models, In the posterior electrodes, there is an overall negative relation of alignment to performance of static models (in all stages but especially I and II), and a positive relation for temporally integrating models. We do not believe we can draw clear conclusions from the specific temporal localization of these findings, however this is consistent with the previous video fMRI work Sartzetaki et al. (2025), also showing no meaningful relationship with task performance across ROIs. This might relate to the arguments in Schrimpf et al. (2018), that beyond a certain task proficiency, improvements in performance do not correlate with improvements in alignment. The task of recognition on ImageNet, e.g. the ability to more accurately discern between two specific dog breeds, is not likely to optimally drive human-like object recognition. Similarly, we also do not find meaningful relationships between alignment and performance in any model type in the frontal electrodes.
> >
> > On top of assessing the relationship between model-brain alignment and model task performance, we also investigate the **relationship between alignment and model computational complexity (flops)**, similar to Sartzetaki et al (2025), for which we see a significant negative relationship in frontal electrodes (Appendix C, Fig. 7 (bottom)) consistent with the relationship in higher level regions, as seen in that study.
> >
> > Schrimpf, M., Kubilius, J., Hong, H., Majaj, N.J., Rajalingham, R., Issa, E.B., Kar, K., Bashivan, P., Prescott-Roy, J., Geiger, F. and Schmidt, K., 2018. Brain-score: Which artificial neural network for object recognition is most brain-like?. BioRxiv, p.407007.

---

### Official Review · Reviewer_mriQ · 2025-11-01

**Soundness:** 3
**Presentation:** 3
**Contribution:** 3
**Rating:** 6
**Confidence:** 2

**Summary:**

The paper presents the first large-scale benchmarking of over 100 deep neural networks against EEG recordings collected during natural video viewing. Using a novel Cross-Temporal Representational Similarity Analysis (CT-RSA), the authors analyze alignment dynamics between model features and brain responses. Results reveal distinct temporal processing stages in posterior and frontal electrodes, suggesting that brain activity during video perception reflects a dynamic mixture of expert models tuned to different representational demands.

**Strengths:**

* Interesting large-scale study linking deep video models with dynamic EEG signals.
* Introduction of CT-RSA as a principled extension of RSA to capture cross-temporal alignment.
* Rich, interpretable findings showing stage-wise specialization in brain responses and model correspondences.
* Strong relevance to both neuroscience and AI, offering biologically inspired insights for model design.

**Weaknesses:**

* While conceptually interesting, the empirical novelty beyond previous fMRI-based benchmarks feels incremental, focusing mainly on EEG substitution.
* The EEG dataset is limited to six subjects, which may restrict generalizability.
* Some claims about "mixture of experts" remain metaphorical rather than computationally substantiated.
* The paper does not analyze the robustness of CT-RSA to preprocessing or electrode partitioning choices, nor quantify its improvement over standard RSA or fMRI-based alignment methods, leaving the methodological advantage insufficiently demonstrated.
* To what extent do the findings generalize across individuals, given the small sample size?

**Questions:**

See above.

---

> ### Author Response · Authors · 2025-11-24
>
> Thank you for your positive comments and constructive feedback. We have revised the manuscript and uploaded the new version with all changes relative to the original shown in blue. Our responses to your points one by one are as follows.
>
> ### W1: While conceptually interesting, the empirical novelty beyond previous fMRI-based benchmarks feels incremental, focusing mainly on EEG substitution.
>
> In our opinion, the realm of fine-grained temporal neural dynamics in the context of videos is hugely underexplored. By using EEG data, we are not simply substituting one measurement modality for another; rather, we gain access to a rich temporal axis that enables entirely new insights and forms of analysis, such as temporal matching of time-unrolled model features to EEG. Implementing this also calls for the introduction of new methods. We propose CT-RSA as a temporal expansion of RSA to handle this temporal matching - this constitutes a methodological novelty of our work. These new forms of analyses have allowed us to uncover new empirical insights that extend beyond the temporal dynamics previously established for static object processing; we see an extended engagement with mid-level temporally integrative action features after the end of the initial low-to-high level cascade of information as previously observed in static image vision, and following initial alignment to high-level action features in frontal electrodes, potentially indicating long-range feedback between cortical areas beyond visual cortex. These findings are also novel and complementary compared to the respective fMRI study (Sartzetaki et al. 2025), where due to the indirect and slow blood flow measurement of fMRI it is impossible to link the time-unrolled model features to sub-second time scale of neural activity, necessary to study the temporal processing of short videos. We have made our contributions relative to these former fMRI results more clear in the revised manuscript.
>
> ### W2: The EEG dataset is limited to six subjects, which may restrict generalizability.
>
> Our set up follows the paradigm of extensive within-subject sampling, similar to the Natural Scenes Dataset (Allen et al., 2022), Bold Moments Dataset (Lahner et al., 2024) and THINGS-EEG (Gifford et al., 2022), providing neural recordings for 1000 videos with 6 repeats for the train set and 102 videos with 24 repeats in the test set. This design is supported by works on scaling laws showing that decodability of brain signals primarily depends on the amount of data per subject, rather than the number of subjects (Banville et al, 2025).
>
> Banville, H., Benchetrit, Y., d'Ascoli, S., Rapin, J., & King, J. R. (2025). Scaling laws for decoding images from brain activity. arXiv preprint arXiv:2501.15322.
>
> Lahner, B., Dwivedi, K., Iamshchinina, P., Graumann, M., Lascelles, A., Roig, G., ... & Cichy, R. (2024). Modeling short visual events through the BOLD moments video fMRI dataset and metadata. Nature communications, 15(1), 6241.
>
> Allen, E. J., St-Yves, G., Wu, Y., Breedlove, J. L., Prince, J. S., Dowdle, L. T., ... & Kay, K. (2022). A massive 7T fMRI dataset to bridge cognitive neuroscience and artificial intelligence. Nature neuroscience, 25(1), 116-126.
>
> Gifford, A. T., Dwivedi, K., Roig, G., & Cichy, R. M. (2022). A large and rich EEG dataset for modeling human visual object recognition. NeuroImage, 264, 119754.

---

> > ### Author Response · Authors · 2025-11-24
> >
> > ### W3: Some claims about "mixture of experts" remain metaphorical rather than computationally substantiated.
> >
> > We agree that the title of the paper is more metaphorical rather than referring to the particular “Mixture of Experts (MoE)” architecture in machine learning. Although the definition of that architecture itself is also quite broad, we purposefully select a slightly different wording which is “(dynamic) mixture of expert models” to refer to the specific expert models that we identified (object, action, static, temporally-integrating) as best matching to different stages of neural processing.
> > However, this play on words with MoEs was also selected so as to encourage (and perhaps provoke) the machine learning and cognitive neuroscience communities to think in a more unified way again. Although not all artificial intelligence architectures need to be inspired by biological intelligence (a debate which made the communities drift apart), similarities in the engineering and biological solutions to tasks can still be found, pointing to core principles that are required for solving a task, regardless of the exact implementation. In the case of MoEs, parallels can be drawn to sparse coding (Olshausen et al., 2004) and they have also recently been used to model established high-level brain processes (Cook et al., 2025, AlKhamissi et al., 2025). Our proposition in this work is that this principle of iteratively engaging different sub-networks in a larger network is  very promising for the task of video understanding (indicated through the interchange of brain alignment between model types), where to the best of our knowledge, it has not yet been explored. We aim to invite debate on how the two communities (in common venues important to both such as ICLR) can now discuss common solutions to tasks on the basis of representational alignment (Sucholutsky et al., 2025) and the iterative model testing cycle (Doerig et al., 2023) that can reveal both new neuroscientific insights through DNN hypothesis testing (e.g. different representational demands across time) and new model designs inspired by those insights, which can make DNNs more efficient and robust similar to the brain (please see response to reviewer 3fFP - W1 for examples of this in recent literature).
> > We recognize that this intention was not sufficiently discussed in the current paper, so we have extended our discussion section.
> >
> > Olshausen, B.A. and Field, D.J., 2004. Sparse coding of sensory inputs. Current opinion in neurobiology, 14(4), pp.481-487.
> >
> > Cook, J., Akarca, D., Costa, R.P. and Achterberg, J., 2025. Brain-Like Processing Pathways Form in Models With Heterogeneous Experts. arXiv preprint arXiv:2506.02813.
> >
> > AlKhamissi, B., De Sabbata, C.N., Tuckute, G., Chen, Z., Schrimpf, M. and Bosselut, A., 2025. Mixture of Cognitive Reasoners: Modular Reasoning with Brain-Like Specialization. arXiv preprint arXiv:2506.13331.
> >
> > Doerig, A., Sommers, R. P., Seeliger, K., Richards, B., Ismael, J., Lindsay, G. W., ... & Kietzmann, T. C. (2023). The neuroconnectionist research programme. Nature Reviews Neuroscience, 24(7), 431-450.
> >
> > Sucholutsky, I., Muttenthaler, L., Weller, A., Peng, A., Bobu, A., Kim, B., … & Griffiths, T. L. (2025). Getting aligned on representational alignment. Transactions on Machine Learning Research (TMLR). Retrieved from https://openreview.net/forum?id=Hiq7lUh4Yn
> >
> > ### W4: The paper does not analyze the robustness of CT-RSA to preprocessing or electrode partitioning choices, nor quantify its improvement over standard RSA or fMRI-based alignment methods, leaving the methodological advantage insufficiently demonstrated.
> > The key advantages of CT-RSA stem from opening up the temporal axis of model representations, providing an additional dimension of analysis, namely the model timepoint, as illustrated in Fig. 2C. Beyond this conceptual extension, CT-RSA provides more accurate alignment by allowing a many-to-many mapping between model and brain time points. As a result, it consistently yields higher alignment scores than standard time-agnostic RSA, which relies on a one-to-many mapping using temporally averaged model features (see Fig. 18 in comparison to Fig. 17). This increase in alignment is especially visible when the brain signal is found to be highly temporal, i.e. here in posterior electrodes. Regarding preprocessing effects, CT-RSA behaves similarly to standard RSA, as it effectively extends a one-to-many RDM comparison into a many-to-many RDM comparison without altering the underlying principles. For a detailed discussion of preprocessing influences, we refer to Guggenmos et al. (2018).
> >
> > Guggenmos, M., Sterzer, P., & Cichy, R. M. (2018). Multivariate pattern analysis for MEG: A comparison of dissimilarity measures. Neuroimage, 173, 434-447.

---

> > > ### Author Response · Authors · 2025-11-24
> > >
> > > ### W5: To what extent do the findings generalize across individuals, given the small sample size?
> > >
> > > Thank you for the question. We have added subject level results to the Appendix C Fig.  We mostly see congruent results with our main analyses on the group-average level, although variations across participants exist. This might be partially due to idiosyncratic differences between subjects and partially due to difference in data quality per subject (see differences in decoding scores over subjects in Appendix C Fig. 6). Therefore, to get the most reliable outcomes we chose to work with the group-averaged data for our main analyses to increase overall the signal-to-noise ratio of the EEG measurements and to focus on variability across models, which was central to our research question given the large number of models included.

---

### Official Review · Reviewer_3fFP · 2025-11-01

**Soundness:** 3
**Presentation:** 3
**Contribution:** 3
**Rating:** 4
**Confidence:** 2

**Summary:**

This study compares deep learning models to brain activity (EEG) and finds that the brain processes videos like a "dynamic mixture of experts." Different brain regions align with different model typesposterior areas with mid-level dynamic features and frontal areas with high-level static actions. The findings suggest that a single, ideal AI model would need to flexibly combine these capabilities.

**Strengths:**

1. The paper is clearly written and well-organized, making it reader-friendly.

2. The article conducted a large number of rigorous experiments and collected new EEG data to explore this issue.

3. The article provides a reasonable explanation for all the results obtained from the experiment, and it is consistent with some existing conclusions in the field of neuroscience and model explanations.

**Weaknesses:**

1. From several articles about brain and model alignment (not limited to video understanding), they all stated that aligning research can help in designing better deep learning models. However, no improved models have been created based on these works. Each time a new model is developed, alignment and explanation are made based on it (e.g., Mixture of Experts, MoE has been used for several years, not based on this article). Even some brain activity explanations of BERT [1] have now been considered outdated by engineering standards. Could the author provide me with some examples of recent studies on brain activity that actually guide the design of deep learning models?

2. It is uncertain how reliable the research on EEG signal alignment is. In my view, EEG signals are very blurry and unstable. They contain a lot of noise, and even for the same subject, EEG responses to the same stimulus can vary when collected multiple times. Did the author first verify the reliability of the collected EEG signals? For example, conducting multiple collections to verify consistency, etc.

3. Furthermore, all the video clips of the experimental studies were only 3 seconds long. It is completely fine to explore the basic visual responses or decode visual signals within this length. But can we really explore the complex reactions of the brain within this time window? (For example, video understanding) Some of the conclusions in the article may have little connection with video understanding. For instance, the initial brain activity of each video segment might merely indicate that we have just viewed a new video. Among the fleeting content, this article not only studies comprehension but also examines brain activity and the changes in the model over time. I'm not sure if this is reliable.

4. Furthermore, current research mainly focuses on alignment, meaning which time points show changes in brain activity. However, we do not understand what these changes actually represent. Although the author provided an explanation, it does not serve as proof. In other words, these changes can be interpreted in various ways, and there is no way to tell which interpretation is correct. It is also unreasonable (no evidence) to assume that the model should match the EEG (brain) activity. In other words, we neither understand how the brain works nor grasp why the model functions. So, does alignment make any sense in this situation?

[1] BERT: Pre-training of Deep Bidirectional Transformers for Language Understanding


Declaration: I lack profound background knowledge of brain-model alignment. But I have interests and questions for this field. If the questions raised are unreasonable or if there are common limitations in the field, the author can provide an explanation. Also, please ACs consider the weight of my questions as appropriate.

**Questions:**

Please see the Weaknesses section.

---

> ### Author Response · Authors · 2025-11-24
>
> Thank you for your review and interest in the field. We have revised the manuscript and uploaded the new version with all changes relative to the original shown in blue. Our responses to your points one by one are as follows.

---

> > ### Author Response · Authors · 2025-11-24
> >
> > ### W1: From several articles about brain and model alignment (not limited to video understanding), they all stated that aligning research can help in designing better deep learning models. However, no improved models have been created based on these works. Each time a new model is developed, alignment and explanation are made based on it (e.g., Mixture of Experts, MoE has been used for several years, not based on this article). Even some brain activity explanations of BERT [1] have now been considered outdated by engineering standards. Could the author provide me with some examples of recent studies on brain activity that actually guide the design of deep learning models?
> >
> > Thank you for raising this important question and for the opportunity to clarify what we believe is a common misconception in recent years; that because engineering in artificial intelligence drifted away from initial inspiration (MLPs, CNNs, RL) from biological intelligence for improved task performance, it means AI has nothing more to gain from neuroscience. This is far from true, especially where models lack more than in performance, which is in robustness, efficiency, shortcut learning (e.g. texture bias, Geirhos et al., 2018), and interpretability. There are many recent examples where models were designed (either in architecture or training regime) with inspiration from neuroscience and showed remarkable advantages in those respects. Topographic neural networks (Lu et al. 2025a, Rathi et al., 2025) operate in a low energy budget and provide interpretability; curriculum learning based on human development (Lu et al. 2025b, Vogelsang et al., 2024a,b) leads to models that are more robust (to image distortions, adversarial attacks) and less prone to shortcut learning of texture biases. In Bagad et al (2025), inspiration from perceptual straightening (Henaff et al., 2019) showed impressive improvement in the time-sensitivity of video models, as well as improvements in action classification performance. In Konkle et al. (2023), implementing brain-like feedback pathways led to improved performance, robustness, recognition in composite images, and prevented hallucinations. For MoEs specifically, very recent work implements a brain inspired language model building on MoEs and specific “experts” found in brain studies, showing increased interpretability (AlKhamissi et al., 2025).
> >
> > All the above works use inspiration from neuroscience to guide the design of models that are better in robustness, efficiency, shortcut learning, and interpretability while maintaining or increasing task performance. However, the neuroscientific insights that lead to these works have been around longer than representational alignment research between brain and DNNs has (there is a lag between discovery, replication & verification, and application). There have been multiple new neuroscientific insights coming from representational alignment in vision (Konkle et al., 2022, Prince et al., 2024, Dwivedi et al., 2021, Graumann et al., 2022, Xie et al., 2020), language (Mahowald et al., 2024, Caucheteux et al., 2021;2023), and multimodal integration (Wang et al., 2023), while as a research programme it has been formally characterized as highly progressive (Doerig et al., 2023). Consequently, many of these insights can in turn be used as inspiration for further model development, often referred to as “closing the loop”; but so far this remains underexplored as the field is quite young. Closing the loop can also be done through enforcing model alignment with neural and behavioral representations explicitly, e.g. through ‘brain-tuning’ or fine-tuning on human behavioral judgments (Safrani et al. 2021, Li et al., 2019, Dapello et al., 2023, Moussa et al., 2025, Muttenthaler et al., 2025, Shao et al., 2024) with improvements in robustness, generalization and performance, without going to the intermediate step of gaining a neuroscientific insight.
> >
> > Thus, evidence from recent works either using neuroscientific insights to guide model design or directly co-training models with human neural and behavioral data, shows that alignment research holds great potential for having better, more efficient, and more robust, computer vision models. We believe our work highlights the interplay machine learning and neuroscience can have in the future (with inspiration going both ways), instead of a divide. Here, through measuring alignment of multiple DNN model types (object, action, static, temporally-integrating) we discover new neuroscientific insights (dynamic interchange of the above “expert” models across time) that point back to existing machine learning architectures such as MoEs as potential candidates to build upon and expand (using principles from the specific brain-alignment patterns we found, from static to dynamic, and from objects to actions) in video understanding where they have not been used before.

---

> > > ### Author Response · Authors · 2025-11-24
> > >
> > > Geirhos, R., Rubisch, P., ..., & Brendel, W., 2018, November. ImageNet-trained CNNs are biased towards texture; increasing shape bias improves accuracy and robustness. In International conference on learning representations (ICLR).
> > >
> > > Konkle, T. and Alvarez, G., 2023. Cognitive steering in deep neural networks via long-range modulatory feedback connections. Advances in Neural Information Processing Systems (NeurIPS), 36, pp.21613-21634.
> > >
> > > Doerig, A., Sommers, R.P., ..., & Kietzmann, T.C., 2023. The neuroconnectionist research programme. Nature Reviews Neuroscience, 24(7), pp.431-450.
> > >
> > > Mahowald, K., Ivanova, A.A., ..., & Fedorenko, E., 2024. Dissociating language and thought in large language models. Trends in cognitive sciences, 28(6), pp.517-540.
> > >
> > > Caucheteux, C., Gramfort, A. and King, J.R., 2021, July. Disentangling syntax and semantics in the brain with deep networks. In International conference on machine learning (pp. 1336-1348). PMLR.
> > >
> > > Caucheteux, C., Gramfort, A. and King, J.R., 2023. Evidence of a predictive coding hierarchy in the human brain listening to speech. Nature human behaviour, 7(3), pp.430-441.
> > >
> > > Konkle, T. and Alvarez, G.A., 2022. A self-supervised domain-general learning framework for human ventral stream representation. Nature communications, 13(1), p.491.
> > >
> > > Prince, J.S., Alvarez, G.A. and Konkle, T., 2024. Contrastive learning explains the emergence and function of visual category-selective regions. Science Advances, 10(39), p.eadl1776.
> > >
> > > Wang, A.Y., Kay, K., Naselaris, T., Tarr, M.J. and Wehbe, L., 2023. Better models of human high-level visual cortex emerge from natural language supervision with a large and diverse dataset. Nature Machine Intelligence, 5(12), pp.1415-1426.
> > >
> > > Dwivedi, K., Bonner, M.F., Cichy, R.M. and Roig, G., 2021. Unveiling functions of the visual cortex using task-specific deep neural networks. PLoS computational biology, 17(8), p.e1009267.
> > >
> > > Graumann, M., Ciuffi, C., Dwivedi, K., Roig, G. and Cichy, R.M., 2022. The spatiotemporal neural dynamics of object location representations in the human brain. Nature human behaviour, 6(6), pp.796-811.
> > >
> > > Xie, S., Kaiser, D. and Cichy, R.M., 2020. Visual imagery and perception share neural representations in the alpha frequency band. Current Biology, 30(13), pp.2621-2627.
> > >
> > > AlKhamissi, B., De Sabbata,..., & Bosselut, A., 2025. Mixture of Cognitive Reasoners: Modular Reasoning with Brain-Like Specialization. arXiv preprint arXiv:2506.13331.
> > >
> > > Hénaff, O.J., Goris, R.L. and Simoncelli, E.P., 2019. Perceptual straightening of natural videos. Nature Neuroscience, 22(6), pp.984-991.
> > >
> > > Bagad, P. and Zisserman, A., 2025. Chirality in Action: Time-Aware Video Representation Learning by Latent Straightening. In The Thirty-ninth Annual Conference on Neural Information Processing Systems (NeurIPS).
> > >
> > > Lu, Z., Thorat, S., Cichy, R.M. and Kietzmann, T.C., 2025. Adopting a human developmental visual diet yields robust, shape-based AI vision. arXiv preprint arXiv:2507.03168.
> > >
> > > Vogelsang, L., Vogelsang, M., Pipa, G., Diamond, S. and Sinha, P., 2024. Butterfly effects in perceptual development: A review of the ‘adaptive initial degradation’hypothesis. Developmental Review, 71, p.101117.
> > >
> > > Vogelsang, M., Vogelsang, L., Pipa, G., Diamond, S. and Sinha, P., Impact of a biomimetic training regimen based on early visual experience on neural network organization and behavior. In NeurIPS 2024 Workshop on Behavioral Machine Learning.
> > >
> > > Rathi, N., Mehrer, J., ..., & Schrimpf, M., 2025. TopoLM: brain-like spatio-functional organization in a topographic language model. The Thirteenth International Conference on Learning Representations (ICLR) 2025.
> > >
> > > Lu, Z., Doerig, A., ..., & Kietzmann, T.C., 2025. End-to-end topographic networks as models of cortical map formation and human visual behaviour. Nature Human Behaviour, pp.1-17.
> > >
> > > Safarani, S., Nix, A., ... & Sinz, F. (2021). Towards robust vision by multi-task learning on monkey visual cortex. Advances in Neural Information Processing Systems (NeurIPS), 34, 739-751.
> > >
> > > Li, Z., Brendel, W., ... & Tolias, A. (2019). Learning from brains how to regularize machines. Advances in neural information processing systems (NeurIPS), 32.
> > >
> > > Dapello, J., Kar, K., ..., & DiCarlo, J. J. (2023). Aligning model and macaque inferior temporal cortex representations improves model-to-human behavioral alignment and adversarial robustness.  In The Eleventh International Conference on Learning Representations (ICLR).
> > >
> > > Shao, Z., Ma, L., ..., & Beck, D. M. (2024). Probing Human Visual Robustness with Neurally-Guided Deep Neural Networks. arXiv preprint arXiv:2405.02564.
> > >
> > > Muttenthaler, L., Greff, K., ... & Lampinen, A. K. (2025). Aligning machine and human visual representations across abstraction levels. Nature, 647(8089), 349-355.
> > >
> > > Moussa, O., Klakow, D., & Toneva, M. (2025). Improving semantic understanding in speech language models via brain-tuning. The Thirteenth International Conference on Learning Representations (ICLR).

---

> > > > ### Author Response · Authors · 2025-11-24
> > > >
> > > > ### W2: It is uncertain how reliable the research on EEG signal alignment is. In my view, EEG signals are very blurry and unstable. They contain a lot of noise, and even for the same subject, EEG responses to the same stimulus can vary when collected multiple times. Did the author first verify the reliability of the collected EEG signals? For example, conducting multiple collections to verify consistency, etc.
> > > >
> > > > To mitigate the influence of noise, we used the high-repetition test set, which includes 24 repetitions of each stimulus per subject, which we average to increase signal to noise ratio. To ensure the EEG signal quality, we have included additional decoding analyses (Appendix C, Fig. 6), which has shown (in the context of image processing) that M/EEG signal can be used to reliably decode stimulus identity and category information (Cichy et al. 2014, Gifford et al. 2022). Our additional decoding analyses show that stimulus identity (i.e., which video was shown to the participant) can be reliably decoded above chance level from the EEG signal in the test set for both the posterior and frontal electrode partitions across all subjects. They also show the importance of using many repetitions per stimulus (Banville et al. 2025) and further motivate our use of the test set for the main results (with 24 reps/stimulus), as videos from the training set (which have 6 reps/stimulus) have much lower decoding accuracy. Lastly, all reported model–brain alignment scores are scaled by the upper noise ceiling (see Methods section). This scaling accounts for inter-subject variability and allows us to interpret alignment relative to the maximum achievable agreement given variability across participants.
> > > >
> > > > Cichy, R. M., Pantazis, D., & Oliva, A. (2014). Resolving human object recognition in space and time. Nature neuroscience, 17(3), 455-462.
> > > >
> > > > Gifford, A. T., Dwivedi, K., Roig, G., & Cichy, R. M. (2022). A large and rich EEG dataset for modeling human visual object recognition. NeuroImage, 264, 119754.
> > > >
> > > > Banville, H., Benchetrit, Y., d'Ascoli, S., Rapin, J., & King, J. R. (2025). Scaling laws for decoding images from brain activity. arXiv preprint arXiv:2501.15322.

---

> > > > > ### Author Response · Authors · 2025-11-24
> > > > >
> > > > > ### W3: Furthermore, all the video clips of the experimental studies were only 3 seconds long. It is completely fine to explore the basic visual responses or decode visual signals within this length. But can we really explore the complex reactions of the brain within this time window? (For example, video understanding) Some of the conclusions in the article may have little connection with video understanding. For instance, the initial brain activity of each video segment might merely indicate that we have just viewed a new video. Among the fleeting content, this article not only studies comprehension but also examines brain activity and the changes in the model over time. I'm not sure if this is reliable.
> > > > >
> > > > > Thank you for the interesting question. Even in fast viewing paradigms of images, semantic understanding of objects has been found to arise as soon as 50 ms (Cichy et al., 2014), and of actions at 250 ms (Zimmermann et al., 2025). If action understanding can occur within 250 ms after viewing single frames, it should be possible to occur within the same duration of video perception. Notably, Karapetian et al. (2025) show that action-related features are processed even slightly faster during video than static frame presentation. Given the limited existing work on the temporal neural dynamics involved in the transition from natural image perception to natural video perception, we intentionally focus on short videos of 3s, which are still more than 6 times longer than this previously measured earliest timing of semantic decodability.  As we observe alignment between high level model features and neural activity within 500ms, within both frontal and posterior activity, our results reinforce that some degree of conceptual processing is already taking place within these short timescales. By further extending to 3s,  we uncover a “return” to mid-level features in posterior activity after ~1 second of neural processing, most aligned for temporally integrating models. We believe this constitutes a novel insight into how “video understanding” happens in the brain. Additionally, in machine learning, action recognition on videos of analogous duration such as Something-Something (2-6 s long), is also called “video understanding”, as even in this timescale it can be non-trivial to differentiate between actions such as the opening and closing of doors. At the same time video models also have a short viewing window due to computational restrictions (up to 32 frames in our case), so future advancements in model context windows would also further enable using longer videos. Finally, extending to intermediate-duration videos (10–20 seconds) is indeed a very exciting direction for future work, as such stimuli introduce additional factors, such as visual surprise, camera motion, and scene cuts, that may reveal new aspects of temporal processing. We have added this suggestion for future research to the discussion.
> > > > >
> > > > > Cichy, R.M., Pantazis, D. and Oliva, A., 2014. Resolving human object recognition in space and time. Nature neuroscience, 17(3), pp.455-462.
> > > > >
> > > > > Zimmermann, M. and Lingnau, A., 2025. The spatiotemporal neural dynamics of action-related features underlying action recognition. Imaging Neuroscience.
> > > > >
> > > > > Bankson, B. B., Hebart, M. N., Groen, I. I., & Baker, C. I. (2018). The temporal evolution of conceptual object representations revealed through models of behavior, semantics and deep neural networks. NeuroImage, 178, 172-182.

---

> > > > > > ### Author Response · Authors · 2025-11-24
> > > > > >
> > > > > > ### W4: Furthermore, current research mainly focuses on alignment, meaning which time points show changes in brain activity. However, we do not understand what these changes actually represent. Although the author provided an explanation, it does not serve as proof. In other words, these changes can be interpreted in various ways, and there is no way to tell which interpretation is correct. It is also unreasonable (no evidence) to assume that the model should match the EEG (brain) activity. In other words, we neither understand how the brain works nor grasp why the model functions. So, does alignment make any sense in this situation?
> > > > > >
> > > > > > Thank you for raising this point. This is a common critique (referred to as black box vs. black box) that was mainly raised at the beginning of this emergent field, which we believe is refuted by several theoretical framework papers (Doerig et al. (2023); Kriegeskorte & Douglas (2018); Kanwisher et al., 2023; Sucholutsky et al., 2025).
> > > > > > Our research follows this framework, in which task-performing models are used to test hypotheses about what computations matter to neural processing in a particular context, based on varying key aspects (e.g. architecture, learning rules, training dataset) one at a time to create a contrastive comparison of representational similarity. While no one claims that these DNN models should replicate brain representations perfectly, they nonetheless show substantial alignment with neural activity, much more than any other computational model of vision ever used before (Cadieu et al., 2014, Khaligh-Razavi et al., 2014).
> > > > > > For more recent works using DNNs for hypothesis testing, see the many insightful works cited in response to W1: for vision (Konkle et al., 2022, Prince et al., 2024, Dwivedi et al., 2021, Graumann et al., 2022, Xie et al., 2020), language (Mahowald et al., 2024, Caucheteux et al., 2021;2023), and multimodal integration (Wang et al., 2023). Along these works, our findings show that certain models gain an advantage at particular temporal stages of video processing, revealing which representational features are most relevant to the brain during those time windows. This model-based dissociation helps us to further build theory about the functionality of the brain.
> > > > > >
> > > > > > Cadieu, C.F., Hong, H., Yamins, D.L., Pinto, N., Ardila, D., Solomon, E.A., Majaj, N.J. and DiCarlo, J.J., 2014. Deep neural networks rival the representation of primate IT cortex for core visual object recognition. PLoS computational biology, 10(12), p.e1003963.
> > > > > >
> > > > > > Khaligh-Razavi, S.M. and Kriegeskorte, N., 2014. Deep supervised, but not unsupervised, models may explain IT cortical representation. PLoS computational biology, 10(11), p.e1003915.
> > > > > >
> > > > > > Doerig, A., Sommers, R. P., Seeliger, K., Richards, B., Ismael, J., Lindsay, G. W., ... & Kietzmann, T. C. (2023). The neuroconnectionist research programme. Nature Reviews Neuroscience, 24(7), 431-450.
> > > > > >
> > > > > > Kriegeskorte, N., & Douglas, P. K. (2018). Cognitive computational neuroscience. Nature neuroscience, 21(9), 1148-1160.
> > > > > >
> > > > > > Kanwisher, N., Khosla, M., & Dobs, K. (2023). Using artificial neural networks to ask ‘why’questions of minds and brains. Trends in Neurosciences, 46(3), 240-254.
> > > > > >
> > > > > > Sucholutsky, I., Muttenthaler, L., Weller, A., Peng, A., Bobu, A., Kim, B., … & Griffiths, T. L. (2025). Getting aligned on representational alignment. Transactions on Machine Learning Research (TMLR). Retrieved from https://openreview.net/forum?id=Hiq7lUh4Yn

---

### Official Review · Reviewer_uQvM · 2025-11-01

**Soundness:** 3
**Presentation:** 3
**Contribution:** 2
**Rating:** 6
**Confidence:** 4

**Summary:**

This paper presents the first large-scale representational alignment benchmarking study between human EEG recordings during natural video viewing and deep neural networks. They benchmark 110 architectures, including imaging models, video models, and state-space model and using a Cross-Temporal Representational Similarity Analysis (CT-RSA), that extends the classic RSA analysis along the temporal dimension. It works by building representational dissimilarity matrices (RDMs) for each EEG timepoints and for each model layer and timepoints across all videos, correlating them to form a cross-temporal map of brain-model correspondence. By extending RSA into the temporal domain, the method aims to reveal when and how neural network representations align with evolving brain activity during continuous visual perception.

**Strengths:**

The CT-RSA framework is an elegant methodology, allowing dynamic model-brain alignment to be explored at high temporal sampling rate using EEG datasets. The paper contributes to bridging fMRI-based alignment work (limited by slow dynamics in general) and fast EEG dynamics, offering complementary insights into temporal processing. The framework is generalisable and could serve as a benchmark pipeline for cross-modal studies comparing other neural modalities (MEG, ECoG, fMRI)

The paper is globally well-written paper, with thorough contextualisation (related work section), and an extensive discussion sections.

The study covers an unusually wide model space (over 110 architectures).

The results uncover a meaningful temporal hierarchy: posterior EEG activity transitions from low-level visual encoding to mid-level integrative representations, while frontal activity remains stable and semantically oriented.

Interesting exploratory results between frontal and posterior electrodes encoding preference w.r.t various types of models (image, video) and training schemes (pre-training, finetuning only)

**Weaknesses:**

Despite the large-scale benchmarking, it looks like the neuroscientific interpretation remains underdeveloped/underexploited:
- The paper provides little theoretical discussion linking the observed dynamics to known cortical hierarchies or cognitive models of video perception.
- It is not clear why this segregation (posterior/frontal) of electrodes was chosen. Was other cluster of electrodes tested? What is the limit of this choice?

Regarding the methods, one could highlight a few blind spots in the methodology:
- The temporal stages (I-IV) are defined by visual inspection rather than a formal statistical segmentation or theoretical motivations, which weakens their interpretability.
- While CT-RSA is a strong method, the heavy reliance on correlation-based RSA prevents causal or mechanistic insights; additional decoding analyses could strengthen the conclusions.
- The methodology does not account for the spatial-temporal organisation of brain activity or the existence of functional networks or prior anatomical organisation; instead, it effectively collapses the EEG pattern at each time step into a single similarity value, losing information about spatial interactions and network-level dynamics.
- It would have been interesting to evaluate differences between subjects (if any?) or mentioned it otherwise.

Results:
- The paper could benefit from a clearer distinction between what is novel and what is confirmatory from the previous fMRI work (Sartzetaki et al. (2025)).
- It is difficult to draw any clear conclusions regarding the results of different type of DL architecture and training schemes, and how it compares to previous work with fMRI.

Limitations of the methods:
- There is not enough discussion of the limits of using EEG for this kind of study (spatial resolution), despite the advantage of the temporal resolution. How do you approach this tradeoff?
- Some part could benefit from more clarity: "Responses in posterior electrodes, after initial alignment to hierarchical static object processing, best align to mid-level representations of temporally-integrative actions and closely match the unfolding video content."
- Figure 1 could be more clear.

**Questions:**

- The alignment of CNNs with frontal EEG activity seems counterintuitive; could the authors expand on this?

- How does correlation metrics evolve with relative models performance of image and video models on there respective benchmarks? Related to that, could the authors clarify how self-supervised models were selected and whether their pretraining objectives influence alignment differently than supervised models?

- How robust are the results to variations in video duration or temporal resolution - for example, would similar dynamics appear with longer or continuous movie stimuli? what do you expect?

- How sensitive is CT-RSA to differences in model frame sampling rates? what would happen if video models were trained on the same sampling rate as the EEG signal is processed?

- Could CT-RSA be extended to multimodal representations (audio-visual or language-conditioned models) to explore cross-modal dynamics in brain alignment? One could think of movie-watching experiments (mostly with fMRI) for this kind of studies as some aspects of cognition can only be unlocked from longer stimuli, especially with complex stimuli such as audio-visual.

- It is not a strong opinion, but the title of the paper might be misleading with respect to the reference to "Mixture of Experts" (MoEs). The “mixture of experts” framing is conceptually interesting but remains speculative; the authors do not provide evidence of explicit modular switching mechanisms.

---

> ### Author Response · Authors · 2025-11-24
>
> Thank you for your constructive and insightful review, appreciation of our contributions, as well as your suggestions, which we believe concretely improve the paper.
> We have revised the manuscript and uploaded the new version with all changes relative to the original shown in blue. Our responses to your points one by one are as follows.
>
> ### W1: The paper provides little theoretical discussion linking the observed dynamics to known cortical hierarchies or cognitive models of video perception.
>
> Thank you for raising this point. The human visual cortex is indeed thought to implement a cortical hierarchy of increasingly complex feature processing, both across cortical space (low-to-high visual areas) and time (early-to-late processing). We indeed do not directly link our observed dynamics to spatial cortical hierarchies (other than making a very coarse distinction between posterior and frontal processing) because of EEG recordings’ limited spatial resolution.. However, the complementary fMRI dataset (using the same videos) analyzed in a prior study (Sartzetaki et al., 2025) does allow us to suggestively relate our findings to the spatial organization found there. For temporal hierarchies, we do directly compare with prior work (discussion section “What we learn by moving to dynamic natural stimuli”) on image perception and video perception studies limited to 1s. We show novel findings that go significantly beyond the known temporal hierarchy of low to high level features,  in the form of a subsequent extended period of alignment mediated by mid-level temporally-integrative action features. This is an exciting new finding that can inform cognitive models of video perception. Moreover, a future direction that would enable concurrent comparison with spatial and temporal hierarchies is EEG-fMRI fusion, which we mention in the ‘limitations and future work’ section of the discussion.
> We have modified our discussion section to more explicitly reflect differences to prior findings in the new version of the manuscript.
>
> ### W2: It is not clear why this segregation (posterior/frontal) of electrodes was chosen. Was other cluster of electrodes tested? What is the limit of this choice?
>
> We have expanded the second paragraph of section 3.3 to better reflect our motivation. First, as our primary interest lies in visual processing, we focused on posterior electrodes, as this mostly covers visual cortex, consistent with previous EEG studies on visual perception (e.g. Xie et al., 2020, Seijdel et al., 2021, Loke et al., 2024). The inclusion of frontal electrodes was an exploratory choice, motivated by the lack of prior knowledge on the localization of visual processing in longer timescales, and specifically chosen to only make a coarse distinction between opposite ends of the brain’s spatial organization, so as to avoid the spatial smearing inherent to EEG signals. For the same reason, we intentionally did not test other intermediate electrode subsets.  Excitingly, we find new results not reported before for short video perception and image perception, namely a temporally defined engagement of responses in frontal electrodes to visual-related tasks like action semantics. This interesting new finding will need to be replicated and further investigated by future work selectively focused on this. Work by Oyarzo et al. (2025) potentially already provides support by showing that the prefrontal cortex is involved in visual processing under complex conditions during image perception.
>
> Oyarzo, P., Singer, J. J., Kar, K., Vidaurre, D., & Cichy, R. M. (2025). Adaptive recruitment of cortex-wide recurrence for visual object recognition. bioRxiv, 2025-10.
>
> Xie, S., Kaiser, D. and Cichy, R.M., 2020. Visual imagery and perception share neural representations in the alpha frequency band. Current Biology, 30(13), pp.2621-2627.
>
> Seijdel, N., Loke, J., Van de Klundert, R., Van der Meer, M., Quispel, E., Van Gaal, S., De Haan, E.H. and Scholte, H.S., 2021. On the necessity of recurrent processing during object recognition: it depends on the need for scene segmentation. Journal of Neuroscience, 41(29), pp.6281-6289.
>
> Loke, J., Seijdel, N., Snoek, L., Sörensen, L.K., van de Klundert, R., van der Meer, M., Quispel, E., Cappaert, N. and Scholte, H.S., 2024. Human visual cortex and deep convolutional neural network care deeply about object background. Journal of Cognitive Neuroscience, 36(3), pp.551-566.

---

> ### Author Response · Authors · 2025-11-24
>
> ### W3: The temporal stages (I-IV) are defined by visual inspection rather than a formal statistical segmentation or theoretical motivations, which weakens their interpretability.
>
> We agree that stages were defined by visual inspection and therefore used with the goal of global description. Importantly, our main conclusions do not rely on these stage boundaries: all statistical analyses underlying Fig. 2 were performed at each time point independently, without using the stage grouping. These results therefore remain fully interpretable regardless of how the stages are defined. Any additional interpretations referring to stages were included to provide an extended coarse global summary of the temporal profile. We do not make claims about formal temporal boundaries between stages, nor do we draw conclusions that depend on such explicit boundaries.
>
> ### W4: While CT-RSA is a strong method, the heavy reliance on correlation-based RSA prevents causal or mechanistic insights; additional decoding analyses could strengthen the conclusions.
>
> Thank you for your suggestion. We have added additional decoding analyses (see Appendix C, Fig. 6) for both electrode partitions for the test set and train set, which shows that video identity can be reliably decoded above chance in all subjects.
>
> ### W5: The methodology does not account for the spatial-temporal organisation of brain activity or the existence of functional networks or prior anatomical organisation; instead, it effectively collapses the EEG pattern at each time step into a single similarity value, losing information about spatial interactions and network-level dynamics.
>
> We agree that the ultimate goal would be to obtain an understanding of the spatio-temporal dynamics of video perception. However, due to the coarse spatial resolution of EEG we believe that, to analyse spatial interactions and network-level dynamics, techniques such as EEG-fMRI fusion should be used rather than depending on fine-grained electrode sets to represent spatial regions, see also our updated discussion (paragraph 'limitations and future work’). However, as currently no other papers to our knowledge have investigated the temporal dynamics of video perception in EEG, we think our work already provides novel insights without explicit spatial interactions, which fuels and further motivates the need for such future studies.
>
> ### W6: It would have been interesting to evaluate differences between subjects (if any?) or mentioned it otherwise.
>
> Thank you for your suggestion. We have now added individual subject level results for the test set to Appendix C, Fig. 9. We mostly see congruent results with our main analyses on the group-average level, although variations across participants exist. This might be partially due to idiosyncratic differences between subjects and partially due to difference in data quality per subject (see also differences in decoding scores over subjects in Appendix C, Fig. 6). Therefore, to get the most reliable outcomes we chose to work with the group-averaged data for our main analyses to increase overall the signal-to-noise ratio of the EEG measurements and to focus on variability across models, which was central to our research question given the large number of models included.
>
> ### W7: The paper could benefit from a clearer distinction between what is novel and what is confirmatory from the previous fMRI work (Sartzetaki et al. (2025)).
>
> We agree that this distinction was not made sufficiently clear in our discussion section. We have updated the paragraph (‘What we learn from using dynamic brain measurements for benchmarking’) to reflect the differences better. To reiterate here, Sartzetaki et al. (2025) reported that mid-level, temporally integrative video-model features align strongly with early visual cortex. In line with this, we observe an advantage for these mid-level features during later processing stages (from ~0.8 s onward) in posterior electrodes. Moreover, Sartzetaki et al. (2025) found strong alignment of high-level action features in high-level visual cortex; we identify a complementary effect in the frontal electrode partition during Stage II (0.24–0.8 s). Thus, the current EEG work *confirms* that both classification task and temporal integration are important factors driving brain alignment, with the *novel* insight that these factors differentiate not only across spatial regions in the visual cortex but also temporal stages of processing. Moreover, only the high-temporal EEG signal allows for a *novel* demonstration that only posterior signals show a link with model feature timing, as measured via model temporal unfolding.

---

> ### Author Response · Authors · 2025-11-24
>
> ### W8: It is difficult to draw any clear conclusions regarding the results of different type of DL architecture and training schemes, and how it compares to previous work with fMRI.
>
> We agree that our conclusions for these factors of variation were less clearly outlined in the paper than classification task and temporal integration, and that their relation to prior fMRI work was not discussed. We have now remedied that in the paper, and also reiterate here.
>
> For **architecture**, the most clear (based on significance) conclusion would be a disadvantage of CNNs and an advantage of SSMs in posterior phases I and II. Considering the differences of the two architectures, and at the same time their differences to Transformers, this might indicate that global attention as well as recurrent processing are useful for aligning to the initial cascade of information in the brain. This advantage appears to be more related to the recurrent processing across time rather than across space (since for static models it does not persist, see Appendix C, Fig. 12). *Relating to previous work with fMRI* (which did not include SSMs), CNNs and Transformers were overall equivalent in fMRI, with one or the other having a small advantage in some ROIs but no clear overall pattern. Here we mostly find the same, apart from posterior stage I which shows a strong advantage for Transformers. However in fMRI there was a pattern with best aligning layers, which we also find in posterior phases II, III, and IV, especially in phase II (see new addition to the discussion section “What we learn from using dynamic brain measurements for benchmarking”).
>
> For **pretraining type**, we hypothesize that the advantage of self-supervised pretraining in the primarily “object processing” stage could relate to self-supervision enabling generalization to other tasks, while the benefit of no pre-training in the temporally-integrative stage may reflect avoiding shortcut learning of unrelated patterns (Byvshev et al., 2022). *Relating to previous work with fMRI*, pretraining type was not explored there, instead training dataset was explored - here we find no difference between training datasets (Appendix C Figs. 14,15).
>
> ### W9: There is not enough discussion of the limits of using EEG for this kind of study (spatial resolution), despite the advantage of the temporal resolution. How do you approach this tradeoff?
>
> Thank you for pointing this out.  We have added further clarification on this trade-off to the manuscript (see ‘limitations and future work’). We see EEG as a complementary measurement to fMRI, with one being able to gain insights where the other cannot. In the context of studying video perception, fMRI is constrained by the sluggish hemodynamic response, which imposes a lower bound on temporal resolution of approximately one second. EEG, in contrast, can capture neural dynamics at the millisecond scale. Rather than emphasizing the spatial limitations of EEG, our work intentionally leverages its temporal precision, providing insights that are inaccessible to fMRI. This makes our approach complementary to existing studies that focus on the spatial mapping of video perception, which are optimally assessed with fMRI. Moreover, our work provides a necessary prerequisite demonstration of meaningful video-related variance in the EEG signal as captured by DNN models, further motivating EEG-fMRI-fusion, as recommended in our discussion section ‘limitations and future work’, to study the spatio-temporal dynamics of video processing.
>
> ### W10: Some part could benefit from more clarity: (1) "Responses in posterior electrodes, after initial alignment to hierarchical static object processing, best align to mid-level representations of temporally-integrative actions and closely match the unfolding video content." (2) Figure 1 could be more clear.
>
> Thank you for pointing these out.
>
> For (1), we modified the sentence to read “After initial alignment to hierarchical static object processing, responses in posterior electrodes best align to mid-level temporally-integrative action features, showing high temporal correspondence to feature timings.” We hope this is more straightforward, but if not, please let us know.
>
> For (2), as Figure 1 contains many components, could you please clarify which part of it is not sufficiently clear?  We are happy to make all suggested changes.

---

> ### Author Response · Authors · 2025-11-24
>
> ### Q1: The alignment of CNNs with frontal EEG activity seems counterintuitive; could the authors expand on this?
>
> We have expanded our discussion of this point, i.e. why we observe alignment of high-level features, particularly from action-classification models, with frontal electrodes, in the revised Discussion section (“What we learn by moving to dynamic natural stimuli”).
> Considering frontal electrodes was an exploratory choice, which revealed a temporally defined engagement period to visual-related tasks not previously reported e for video perception or image perception. Aside from finding this exciting by itself,  these findings connect to prior work on feedback processes. During object recognition in static contexts, feedback from frontal areas to posterior visual regions plays a critical role in shaping behaviorally sufficient object representations, especially under challenging conditions (Goddard et al., 2016; Kar & DiCarlo, 2021; Oyarzo et al., 2025). Specifically, Oyarzo et al. (2025) pose that the frontal cortex plays a corrective role under these conditions, reshaping high-level visual representations to resolve ambiguity. If we extrapolate, video processing can be viewed as a more demanding extension of static image processing. This increased complexity may explain the recruitment of frontal activity, which appears to contribute early on. Subsequently, a reconfiguration of representations in posterior activity follows, supporting updated mid-level feature processing over time. We note, however, that this is a newly formed hypothesis generated by our findings, that needs to be tested and refined in future studies.
> With regards to a difference between CNNs and Transformers, the advantage is very slight, but it could relate to the best aligned layer being much later (closer to classification) in CNNs, so that would correspond to more semantic meaning and higher level processing.
>
> Oyarzo, P., Singer, J. J., Kar, K., Vidaurre, D., & Cichy, R. M. (2025). Adaptive recruitment of cortex-wide recurrence for visual object recognition. bioRxiv, 2025-10.
>
> Kohitij Kar and James J DiCarlo. Fast recurrent processing via ventrolateral prefrontal cortex is needed by the primate ventral stream for robust core visual object recognition. Neuron, 109(1): 164–176, 2021.
>
> Erin Goddard, Thomas A Carlson, Nadene Dermody, and Alexandra Woolgar. Representational dynamics of object recognition: Feedforward and feedback information flows. Neuroimage, 128: 385–397, 2016.
>
> ### Q2a: How does correlation metrics evolve with relative models performance of image and video models on there respective benchmarks?
>
> Thank you for the question. We provide additional analyses in Appendix C, Fig. 7 (top) that show the **relationship of the model-brain alignment metric to model task performance**, in Kinetics-400 for the action recognition models and in Imagenet-1k for the object recognition models. In the posterior electrodes, there is an overall  negative relation of alignment to performance of static models (in all stages but especially I and II), and a positive relation for temporally integrating models. We do not believe we can draw clear conclusions from the specific temporal localization of these findings, however this is consistent with the previous video fMRI work Sartzetaki et al. (2025), also showing no meaningful relationship with task performance across ROIs. This might relate to the arguments in Schrimpf et al. (2018),  that beyond a certain task proficiency, improvements in performance do not correlate with improvements in alignment any more. The task of recognition on ImageNet, e.g. the ability to more accurately discern between two specific dog breeds, is not likely to optimally drive human-like object recognition. Similarly, we also do not find meaningful relationships between alignment and performance in any model type in the frontal electrodes.
>
> On top of assessing the relationship between model-brain alignment and model task performance, we also investigate the **relationship between alignment and model computational complexity (flops)**, similar to Sartzetaki et al. (2025), for which we see a significant negative relationship in frontal electrodes (Appendix C, Fig. 7 (bottom)) consistent with the relationship in higher level regions as seen in that study.
>
> Schrimpf, M., Kubilius, J., Hong, H., Majaj, N.J., Rajalingham, R., Issa, E.B., Kar, K., Bashivan, P., Prescott-Roy, J., Geiger, F. and Schmidt, K., 2018. Brain-score: Which artificial neural network for object recognition is most brain-like?. BioRxiv, p.407007.

---

> ### Author Response · Authors · 2025-11-24
>
> ### Q2b: Related to that, could the authors clarify how self-supervised models were selected and whether their pretraining objectives influence alignment differently than supervised models?
>
> The selection of self-supervised models was based on a combination of criteria, being state of the art and publicly available models (available in the mmaction library plus self-supervised-pretrained versions of videomamba). Two types of analyses on the influence of self supervision can be envisioned. The first is how pretraining in a self-supervised manner influences alignment (before supervised finetuning), which we show in our pretraining type analysis, main figure 5. This shows that models pretrained with SSL exhibit an advantage in stage II, which we relate to their ability to learn generalizable features that are not task specific (through their objective of reconstructing masked spatiotemporal patches), since at that stage object processing is also important for alignment. The second type of analysis is to compare alignment of purely self-supervised models (without supervised fine-tuning) with that of models having some form of supervision (either from scratch or as finetuning). As we only had one purely self-supervised model available, this analysis is shown in Appendix C, Fig. 16. This figure shows that the purely self-supervised variant of videomamba (without any supervised finetuning) is outperforming variants with supervised training in posterior stages I and II, and frontal stages I and II, but not in posterior stages III and IV.
>
> ### Q3: How robust are the results to variations in video duration or temporal resolution - for example, would similar dynamics appear with longer or continuous movie stimuli? what do you expect?
>
> These are very interesting questions that also constitute exciting directions for future work, with the collection of new neural datasets of intermediate duration videos (10-20s) that could include elements of visual surprise, camera changes, or even scene cuts. We updated our future work section addressing this point. We can hypothesize that the observed dynamics of the current study would replicate as long as most elements of the scene remain stable. As a consequence of significant low-level visual content changes (e.g. visual surprise, camera change, or more extremely a scene cut), we would expect our observed neural computational cascade, i.e. sequential phases of hierarchical object processing and processing of mid-level temporally integrative actions, to start anew, depending on the severity of the change in visual and semantic content. When low-level visual content remains relatively stable we would expect visual suppression (Benda, 2021) to take place, leading to less strong initial visual responses than in the beginning of the video.
> Regarding temporal resolution, we refer you to our answer for your next question.
>
> Benda, J. (2021). Neural adaptation. Current Biology, 31(3), R110-R116.
>
> ### Q4: How sensitive is CT-RSA to differences in model frame sampling rates? what would happen if video models were trained on the same sampling rate as the EEG signal is processed?
>
> Thank you for the interesting question. We provide new analyses to answer this first question in Appendix C, Fig. 8 and also add a results section (“Context window and sampling rate influence alignment in late posterior processing.“) on this aspect. Specifically, we correlate the alignment score with the different frame sampling rates used in the models and find that the models that perform temporal integration benefit from higher sampling rates (in the stages that they provide an alignment advantage, i.e. posterior stages III and IV) , as well as from longer context windows.
> Regarding your second question, these results hint that increasing the sampling rate (to match the temporal resolution of the EEG signal) of these temporally integrative models, could potentially lead to an even better match in alignment in posterior stage III and IV.
> However, it would not be straightforward to obtain a model that leverages the same high sampling rate as the temporal resolution of the EEG, i.e 50 Hz, as the public models available to our knowledge do not provide this (the model within our subset with the highest sampling rate uses 16 fps) and this would be very computationally expensive.
> On the other hand it would be possible to downsample the EEG data to a lower temporal frequency that matches the model sampling rate, to investigate this question. However, doing so would lead to an (unnecessary) loss of potentially valuable information in the brain signal.

---

> ### Author Response · Authors · 2025-11-24
>
> ### Q5: Could CT-RSA be extended to multimodal representations (audio-visual or language-conditioned models) to explore cross-modal dynamics in brain alignment? One could think of movie-watching experiments (mostly with fMRI) for this kind of studies as some aspects of cognition can only be unlocked from longer stimuli, especially with complex stimuli such as audio-visual.
>
> This is an excellent suggestion and indeed a strong potential use case for CT-RSA. Our setup generalizes to any setting that involves comparing two forms of temporal, or sequential, representations. On the model side, this can be multimodal audio-visual and language representations (d’Ascoli et al., 2025, Gifford et al., 2024), either from separate unimodal DNNs or multimodal DNNs, with the inclusion of both enabling the study of (spatio-)temporal hierarchies for multimodal integration (Wang et al., 2023). On the neural side, it could extend to long-format fMRI time series or other sequential neural measurements. For example in the setup of d’Ascoli et al. (2025), encoding could be replaced with CT-RSA as an option for exploring the inherent model-brain alignment instead of maximizing neural predictivity.
> We have added a new section to the Discussion.
>
> d'Ascoli, S., Rapin, J., Benchetrit, Y., Banville, H. and King, J.R., 2025. TRIBE: TRImodal Brain Encoder for whole-brain fMRI response prediction. arXiv preprint arXiv:2507.22229.
>
> Gifford, A.T., Bersch, D., St-Laurent, M., Pinsard, B., Boyle, J., Bellec, L., Oliva, A., Roig, G. and Cichy, R.M., 2024. The algonauts project 2025 challenge: How the human brain makes sense of multimodal movies. arXiv preprint arXiv:2501.00504.
>
> Wang, A.Y., Kay, K., Naselaris, T., Tarr, M.J. and Wehbe, L., 2023. Better models of human high-level visual cortex emerge from natural language supervision with a large and diverse dataset. Nature Machine Intelligence, 5(12), pp.1415-1426.
>
> ### Q6: It is not a strong opinion, but the title of the paper might be misleading with respect to the reference to "Mixture of Experts" (MoEs). The “mixture of experts” framing is conceptually interesting but remains speculative; the authors do not provide evidence of explicit modular switching mechanisms
>
> We agree that the title of the paper is more metaphorical rather than referring to the particular “Mixture of Experts (MoE)” architecture in machine learning. Although the definition of that architecture itself is also quite broad, we purposefully select a slightly different wording which is “(dynamic) mixture of expert models” to refer to the specific expert models that we identified (object, action, static, temporally-integrating) as best matching to different stages of neural processing.
> However, this play on words with MoEs was also selected to encourage (and perhaps provoke) the machine learning and cognitive neuroscience communities to think in a more unified way again. Although not all artificial intelligence architectures need to be inspired by biological intelligence (a debate which made the communities drift apart), similarities in the engineering and biological solutions to tasks can still be found, pointing to core principles that are required for solving a task, regardless of the exact implementation. In the case of MoEs, parallels can be drawn to sparse coding (Olshausen et al., 2004) and they have also recently been used to model established high-level brain processes (Cook et al., 2025, AlKhamissi et al., 2025). Our proposition in this work is that this principle of iteratively engaging different sub-networks in a larger network is very promising for the task of video understanding (indicated through the interchange of brain alignment between model types), where to the best of our knowledge, it has not yet been explored. We aim to invite debate on how the two communities (in common venues important to both such as ICLR) can now discuss common solutions to tasks on the basis of representational alignment (Sucholutsky et al., 2025) and the iterative model testing cycle (Doerig et al., 2023) that can reveal both new neuroscientific insights through DNN hypothesis testing (e.g. different representational demands across time) and new model designs inspired by those insights, which can make DNNs more efficient and robust, similar to the brain (see response to reviewer 3fFP - W1 for more examples of this).
> We recognize that this intention was not sufficiently discussed in the paper, so we have extended our discussion section.
>
> Olshausen, B.A. and Field, D.J., 2004. Sparse coding of sensory inputs. Current opinion in neurobiology, 14(4), pp.481-487.
>
> Cook, J., Akarca, D., Costa, R.P. and Achterberg, J., 2025. Brain-Like Processing Pathways Form in Models With Heterogeneous Experts. arXiv preprint arXiv:2506.02813.
>
> AlKhamissi, B., De Sabbata, C.N., Tuckute, G., Chen, Z., Schrimpf, M. and Bosselut, A., 2025. Mixture of Cognitive Reasoners: Modular Reasoning with Brain-Like Specialization. arXiv preprint arXiv:2506.13331.

---

### Author Response · Authors · 2025-12-02
**Summary of reviews and rebuttals**

Dear reviewers and AC,

We sincerely appreciate your hard work during the review process, especially under these unfortunate and challenging conditions. We were happy to see that the reviewers appreciated the value of our novel proposed method **(uQvM, mriQ)**, the thoroughness and large scale of our experimental design **(uQvM, 3fFP, mriQ)**, novelty of neuroscientific findings **(uQvM, mriQ, fzAq)**, and high relevance to both neuroscience and AI **(mriQ, fzAq)**, as well as the clarity and overall presentation and organization of the paper **(uQvM, 3fFP, fzAq)**. Reviewers also had several suggestions and questions - we believe that the incorporation of this feedback has further strengthened the paper. Below, we provide a high-level summary of the key points addressed in the reviews and rebuttals. We hope this will facilitate you in the final assessment.

### Usage of EEG
There were several questions around the usage of EEG, by reviewer **uQvM** regarding its limited spatial resolution **(W9, W1, W2, W5)** and by reviewer **3fFP** regarding reliability of the signal quality **(W2)**. Based on the suggestion of reviewer **uQvM (W9)**, we have extended our discussion (in “Limitations and future work”) on the trade-offs of EEG, i.e. its high temporal resolution, which we leverage to align time-unrolled model features with temporal neural representations, and its limited spatial resolution, along with directions for future work **(W1, W5 uQvM)**. In addition, we have expanded the methodological description of our electrode selection **(W2 uQvM)** and included additional stimulus identity decoding analyses across all subjects to demonstrate reliable EEG signal quality **(W2 3fFP; W4 uQvM)** *(Fig. 6)*.

### Duration of video stimuli
Reviewers **uQvM (Q3)** and **3fFP (W3)** raised questions regarding the duration of the video stimuli used (3s). With respect to the brain’s ability to achieve high-level understanding within this duration **(W3 3fFP)**, we first provide evidence from prior work that even in under 1s, high-level semantic information of actions is decodable from brain activity. We further highlight that extending to 3s of continuous visual stimulus in our work allows us to uncover new insights, making it an important milestone towards longer stimuli. Subsequently, we discuss potential differences of longer videos (e.g. more camera changes) and our expectations of how our findings might generalize there **(Q3 uQvM; W3 3fFP)**, adding a direction in “Limitations and future work”.

### More extensive interpretation of results
Reviewers requested that we extend our interpretations of our results regarding (1) comparisons to fMRI **(W7, W8 uQvM)**, (2) the pretraining and architecture factors **(W8 uQvM; W2 fzAq)**, and (3) the observed engagement in frontal electrodes **(Q1 uQvM)**.
For (1) we have modified our Discussion to make differences more clear and extend the comparisons to all factors of variation. For (2) we have modified the respective paragraphs in Results concerning architecture and pretraining (last 2 paragraphs). For (3) we extended our Discussion with regards to why we observe alignment of high-level features with frontal electrodes.

### Title and high-level interpretation
In response to reviewers **uQvM (Q6)** and **mriQ (W3)** we clarified that our title does not refer to the particular “Mixture of Experts (MoE)” architecture, but rather to a more conceptual notion of different model expertise (either in classification task or temporal integration) best aligning to brain responses across time. At the same time we intentionally want to hint at prospects for model building, as we believe that this dynamic switching revealed in our results could be key for the efficient processing of videos in humans.

### Background to the field of Neuro-AI
Reviewer **3fFP (W1, W4)** asked for more Neuro-AI background, regarding both its applications for neuro-inspired AI **(W1)** and neural hypothesis testing with AI models used as hypotheses **(W4)**.
In **W1** they requested recent examples of neuro-inspired design of deep learning models, which we provided in addition to discussing the relation to brain-DNN alignment research and to our work. We also enhance the Introduction of our paper with such examples. **W4** relates to a known debate that asks how we can aim to understand DNNs or the brain from comparing their representations, if both of them are “black boxes”, which we also answer based on prior literature in the field.

### Additional analyses
- Relation of brain alignment to model task performance **(Q2a uQvM; Q1 fzAq)** *(Fig. 7)*
- Relation of brain alignment to model sampling frequency and context window length **(Q4 uQvM)** *(Fig. 8)*
- Subject-level brain alignment **(W6 uQvM; W5 mriQ)** *(Fig.9)*

We again thank everyone for their contributions to improving this work. We believe the feedback we received has resulted in a stronger manuscript that better highlights our results.

---

### Meta-Review · Area_Chair_31Wv · 2025-12-18

**Summary:**

This paper introduces a large-scale benchmarking study comparing 110+ deep neural network models with human EEG recordings during natural video viewing. The authors propose Cross-Temporal Representational Similarity Analysis (CT-RSA), an extension of RSA that captures temporal alignment between time-unfolded model features and dynamic brain responses. The study shows that posterior electrodes exhibit a temporal progression from hierarchical object processing to mid-level temporally-integrative action features, while frontal electrodes align with high-level static action representations without temporal correspondence. The authors frame these findings as evidence for "dynamic mixture of expert models" in the brain.

The main concerns raised by reviewers included:
1. spatial resolution limitations of EEG and methodological justification,
1. generalizability with only 6 subjects,
1. short video duration (3s),
1. limited interpretation of some findings (architecture/pretraining effects),
1. incremental novelty beyond fMRI work,
1. reliability of EEG signals, and
1. fundamental questions about the value of alignment research.

The authors provided a balanced summary of the reviews/rebuttal state in their final comment.

**Reviewer Concerns:**

**Addressed:**

* EEG signal reliability (3fFP, uQvM): Authors added decoding analyses (Fig. 6) demonstrating above-chance stimulus identity decoding across all subjects, validating signal quality
* Limited interpretation of results (uQvM, fzAq): Authors expanded the discussion sections, clarified differences from fMRI work and provided deeper interpretation of architecture (SSMs) and pretraining effects
* Missing analyses (uQvM, fzAq): Authors added new analyses relating alignment to task performance, computational complexity, sampling frequency, context window length, and subject-level results (Figs. 7-9)
* Methodological clarity (uQvM, fzAq): Authors expanded explanation of electrode selection rationale and clarified methodological choices regarding Spearman correlation and noise ceiling scaling
* EEG spatial resolution limitations (uQvM): Authors expanded the discussion of EEG/fMRI tradeoffs and limitations, positioning EEG as complementary to fMRI and suggesting EEG-fMRI fusion for future work.
* Video duration concerns (uQvM, 3fFP): Authors provided evidence that semantic understanding occurs within sub-second timescales and justified the 3s duration as an important milestone, but acknowledge this as a direction for future work with longer stimuli
* "Mixture of experts" framing (uQvM, mriQ): Authors clarified this is metaphorical rather than referring to the MoE architecture, and expanded discussion of implications for model design.
* Fundamental value of alignment research (3fFP): Authors provided extensive literature support for the neuroconnectionist framework and examples of neuro-inspired AI

**Partially addressed:**

* Limited sample size (mriQ): While authors justify the within-subject design following established paradigms and show subject-level results (Fig. 9), generalizability across populations still remains factually limited with n=6

**Reviewer Scores:**

* Reviewer uQvM (current: 6): Will likely maintain score of 6 or potentially increase to 7. The reviewer appreciated the large-scale effort and methodology from the start, and the authors addressed all major concerns with substantial new analyses and expanded discussion
* Reviewer 3fFP (current: 4): Could potentially increase to 5-6. The reviewer acknowledged limited domain expertise but raised important questions. The authors provided responses with extensive literature support addressing fundamental concerns about the field's value and EEG reliability
* Reviewer mriQ (current: 6): Will likely maintain score of 6. The reviewer's main concerns about novelty and sample size were addressed with reasonable justifications, though the reviewer could still stay skeptical.
* Reviewer fzAq (current: 4): Could potentially increase to 5-6. The reviewer's concerns about metric justification and interpretation were directly addressed with explanations and expanded discussion

Taken together, I believe this paper would have been accepted in the end.

---

### Decision · Program_Chairs · 2026-01-26

Accept (Poster)